# Two-timescale response of a large Antarctic ice shelf to climate change

Kaitlin A. Naughten [1✉], Jan De Rydt [2], Sebastian H. R. Rosier [2], Adrian Jenkins [2], Paul R. Holland[1] & Jeff K. Ridley[3]

A potentially irreversible threshold in Antarctic ice shelf melting would be crossed if the ocean cavity beneath the large Filchner–Ronne Ice Shelf were to become flooded with warm water from the deep ocean. Previous studies have identified this possibility, but there is great uncertainty as to how easily it could occur. Here, we show, using a coupled ice sheet-ocean model forced by climate change scenarios, that any increase in ice shelf melting is likely to be preceded by an extended period of reduced melting. Climate change weakens the circulation beneath the ice shelf, leading to colder water and reduced melting. Warm water begins to intrude into the cavity when global mean surface temperatures rise by approximately 7 °C above pre-industrial, which is unlikely to occur this century. However, this result should not be considered evidence that the region is unconditionally stable. Unless global temperatures plateau, increased melting will eventually prevail.

[1] British Antarctic Survey, Cambridge, UK. [2] Department of Geography and Environmental Sciences, Northumbria University, Newcastle upon Tyne, UK. [3] Met Office, Exeter, UK. ✉email: kaight@bas.ac.uk

I n order to constrain long-term projections of sea-level rise, it is crucial to understand the likely response of the Filchner–Ronne Ice Shelf (FRIS) to climate change. FRIS is one of Antarctica's largest and most dynamically important ice shelves, which accounts for over 10% of the discharge of the entire Antarctic Ice Sheet[1] and plays a crucial role in global thermohaline circulation[2]. Unlike ice shelves in the Amundsen Sea, which are already in contact with relatively warm water flowing from the deep ocean (≈1 °C)[3], the cavity beneath FRIS is currently filled with cold water (≈−2 °C)[2]. This results in relatively low ice shelf basal melt rates with some regions of refreezing. Whether the FRIS cavity remains in this cold state, or transitions to a warm state akin to the Amundsen Sea, will have significant implications for its source glaciers and ultimately global sea level rise[4].

Circulation beneath FRIS[2] is driven by density gradients between saltier, denser water on the surrounding continental shelf and fresher, lighter water within the cavity. Cold and dense High Salinity Shelf Water (HSSW) flows into the cavity from the strong sea ice formation regions of the Ronne Depression and Berkner Bank (Fig. 1a), where salinity is enhanced by brine rejection. As it travels, the HSSW melts the base of the ice shelf, becoming colder and fresher as meltwater is entrained. The resulting Ice Shelf Water (ISW), supercooled relative to the surface freezing point, exits the cavity through the Filchner Trough.

For decades, it was thought that FRIS melting would decrease with climate change. Observations show that FRIS melt rates are highest when HSSW flow through the cavity is strongest[5]. Nicholls[6] hypothesised that climate warming would provoke a response similar to springtime warming, whereby sea ice formation decreases, the flow of HSSW weakens, and melt rates are reduced.

More recently, Hellmer et al.[7] and related studies[8–10] proposed an alternative hypothesis: that FRIS melting would increase with climate change, triggered by a fundamental change in cavity circulation. Using an ice-ocean model forced with a climate change projection, Warm Deep Water (WDW) from offshore was simulated to enter the cavity through the Filchner Trough, dramatically increasing basal melt rates. A similar circulation change has been triggered in other ocean models by manually varying the cavity salinity or atmospheric forcing[11,12]. However, it remains unclear how easily this circulation change could occur in reality, and over what timescales.

These two conflicting hypotheses for the future of FRIS melting— whether it will decrease or increase—have never been reconciled. Recent advances in Antarctic modelling warrant a re-examination of this topic. All previous simulations projecting WDW intrusion in the 21st century were forced by the same climate model projection, which is now a decade old[13]. In the years since, global climate models have reduced their atmospheric biases in the Antarctic region[14], which increases confidence in their projections. Regional ice-ocean models, which can downscale these results to ice shelf cavities, have become more sophisticated[15] with higher resolution, coupled ice sheet dynamics, and reduced present-day biases. Even so, future projections of ice shelf melting are scarce in the scientific literature.

In this study, we find a two-timescale response of the FRIS cavity to climate warming (Fig. 1b, c). The first stage is an extended period of weakened circulation and reduced basal melting. In the second stage, FRIS melting increases as WDW flows into the cavity via the Filchner Trough. These two stages roughly correspond to the contrasting paradigms of Nicholls[6] and Hellmer et al.[7], showing they are two sides of the same coin. However, the second stage requires extremely strong greenhouse forcing in our simulations, indicating that FRIS melting is unlikely to increase in the 21st century.

## Results

**Experimental design**. Here we present simulations with the coupled ice sheet-ocean model ÚaMITgcm, which simulates ocean circulation, sea ice processes, ice shelf basal melting, and ice sheet flow in the Weddell Sea and FRIS region (Methods). It is forced in the atmosphere and at the lateral ocean boundaries with output from the global climate model UKESM (United Kingdom Earth System Model)[16]. Present-day simulations with this combination of models produce basal melt rates and water mass characteristics exhibiting good agreement with observations, with minimal grounding line drift (Supplementary Note 2).

We simulate three idealised scenarios following the CMIP6 protocol (Coupled Model Intercomparison Project phase 6)[17]: piControl holds $CO_2$ fixed at pre-industrial levels, abrupt-4xCO2 instantaneously quadruples $CO_2$ and then holds it fixed, and 1pctCO2 increases $CO_2$ by 1% per year. All simulations run for 150 years. To investigate longer-term impacts, we then run 50-year extensions of each simulation, where the last decade of UKESM forcing is repeated 5 times. These idealised scenarios were chosen over realistic climate projections, which are shorter with less extreme changes in $CO_2$[18], and may not induce sufficient warming to trigger a circulation change beneath FRIS.

The computational expense of ÚaMITgcm means that a large ensemble of long simulations would be impractical. However, the signal-to-noise ratio of forced change to internal climate variability is relatively small in scenarios with such extreme $CO_2$ forcing. We therefore use a single ensemble member for each UKESM scenario. UKESM was chosen from CMIP6 due to early data availability, as well as its good representation of the present-day Antarctic region[14].

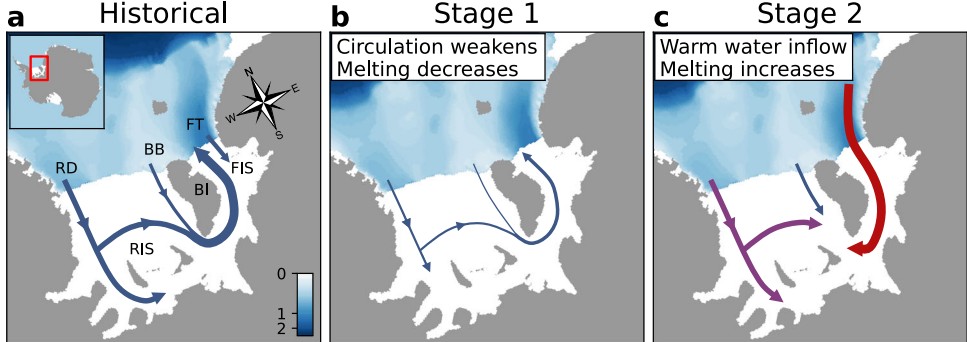

**Fig. 1 Schematic of projected circulation changes in the Filchner–Ronne Ice Shelf cavity. a** Historical conditions; **b** Stage 1 of the response; **c** Stage 2 of the response. Blue arrows show cold High Salinity Shelf Water, red arrows show Warm Deep Water, and purple arrows show Modified Warm Deep Water. The thickness of the arrows indicates the strength of circulation. The colour scale shows bathymetry (seafloor depth) in km. Geographical features are labelled in (**a**) as follows: RD Ronne Depression, BB Berkner Bank, FT Filchner Trough, BI Berkner Island, RIS Ronne Ice Shelf, FIS Filchner Ice Shelf.

**Freshening of continental shelf**. Both stages of the response are ultimately driven by freshening of the source waters which feed the FRIS cavity. Average salinity on the continental shelf in front of FRIS (Fig. 2) decreases by approximately 0.7 psu in 1pctCO2 and almost 1 psu in abrupt-4xCO2, relative to the piControl

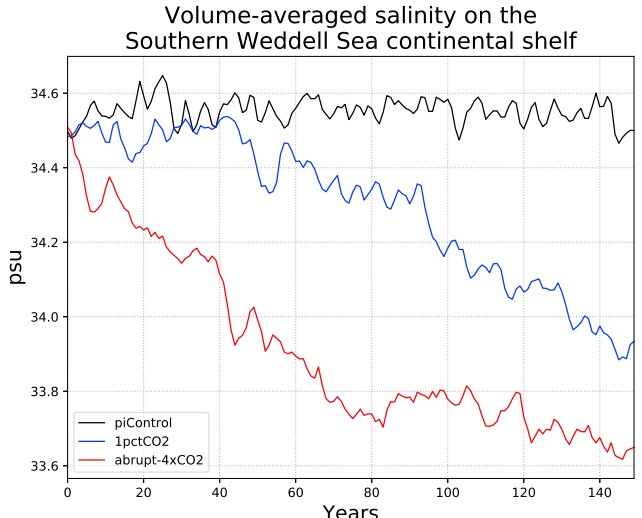

**Fig. 2 Timeseries of salinity over the Southern Weddell Sea continental shelf.** Results are shown annually-averaged for each simulation, and the region is defined in Supplementary Fig. 5.

mean. While the rate of freshening is relatively steady throughout 1pctCO2, in abrupt-4xCO2 it largely plateaus after 80 years, in line with global temperature change.

This freshening is caused by a combination of local and remote factors, which we examine using a salt budget analysis for the abrupt-4xCO2 simulation. Integrated salt fluxes over the continental shelf in front of FRIS (Fig. 3a) show that surface processes are the main cause of the freshening. Surface salt flux anomalies are strongly negative during the entire simulation.

Figure 3b further decomposes the surface processes, showing anomalies in surface freshwater fluxes (note the opposite sign to Fig. 3a, which shows salt fluxes). The increased freshwater is primarily the result of increased sea ice melting. A warmer atmosphere means more sea ice is melting locally, rather than being exported from the continental shelf and later melting offshore. Sea ice freezing initially strengthens (more negative freshwater flux) owing to the greater expanse of open water at the start of each freeze season; however, it only partially compensates for the increased melting. Towards the end of the simulation, sea ice freezing weakens (more positive freshwater flux) as the length of the freeze season becomes shorter. There is a very small contribution from increased precipitation over the continental shelf.

While surface processes dominate, Fig. 3a also shows a freshening contribution from advection beginning around year 40, which indicates that fresher water is being advected into the region along the upstream coastal current. This fresher coastal current is partially caused by increased ice shelf melting in the upstream Eastern Weddell sector, within the ÚaMITgcm domain (eastern boundary 30°E). It also reflects freshening in UKESM,

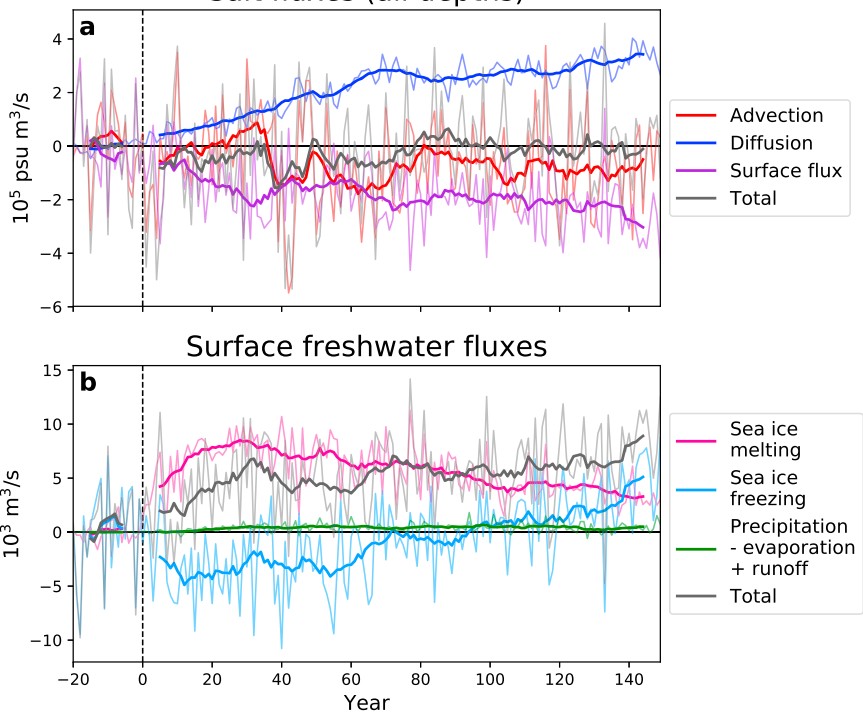

**Fig. 3 Salt budget analysis for the southern Weddell Sea continental shelf.** The timeseries show abrupt-4xCO2 anomalies from the last 20 years of the pre-industrial spinup. The spinup period is shown in the time before year 0, to indicate the variability. Thinner lines show annual averages, while thicker lines show the 11-year running means (calculated separately for the periods before and after year 0 so as not to confound the transition). The region is defined in Supplementary Fig. 5. **a** Integrated salt fluxes in $10^5$ psu m$^3$/s. The advection timeseries includes the surface correction term from MITgcm's linear free surface scheme. **b** Integrated surface freshwater fluxes in $10^3$ m$^3$/s. Note the opposite sign to (**a**). Sea ice freezing refers to the freshwater flux from sea ice freezing, which has the opposite sign to the freezing rate itself.

 3

which then enters the ÚaMITgcm domain at the eastern boundary. The Southern Ocean freshening seen in UKESM is a common response to climate warming among CMIP models[19] and is primarily caused by increased precipitation, with a smaller role for surface melting of the Antarctic Ice Sheet. Precipitation over the Antarctic Ice Sheet in UKESM is not immediately routed to the ocean, but rather is handled by a snow melting scheme. Note that the relatively low resolution of UKESM may decrease confidence in its ability to accurately simulate coastal dynamics[20]. Therefore, some uncertainty exists in the relative role of advection in this salt budget, which could be important for FRIS melt rates[21].

The changes in surface fluxes and advection are partially offset by increased diffusion. After year 80, a new equilibrium is reached and the total salt flux anomalies oscillate around 0, indicating the freshening has plateaued (as in Fig. 2).

In summary, the main causes of freshening are as follows:

1. An increase in local sea ice melting
2. The advection of fresher water into the region (caused by increased ice shelf melting upstream, and increased precipitation over the Southern Ocean)
3. A decrease in sea ice formation, but only in the later stages of the simulation

These mechanisms differ from the hypothesis of Nicholls[6], which relied entirely on a reduction in local sea-ice formation.

**Changes in density gradients**. Continental shelf freshening has implications for the density gradients which drive circulation beneath FRIS. There are two density gradients to consider:

1. between the dense source HSSW (in Ronne Depression and over Berkner Bank) and the less dense FRIS cavity; and
2. between the dense outflowing ISW (in Filchner Trough) and the less dense offshore WDW.

A reversal of the first density gradient characterises Stage 1 of the response, when HSSW inflow slows and ice shelf melting decreases. A reversal of the second density gradient characterises Stage 2, when WDW flows into the cavity and ice shelf melting increases.

Figure 4 illustrates how these density gradients change as the climate warms and the continental shelf freshens. The complexity of ocean circulation means that individual density transects are not simple determinants of the direction of flow, which also depends on topography, wind stress, and many other factors. However, changes in density gradients do influence, and can reasonably explain, circulation changes seen in the simulation (see Supplementary Movie 1).

In the Ronne Depression (Fig. 4a), bottom density is initially relatively constant along the entire length of the depression. HSSW flows freely downslope, southward from the continental shelf into the western Ronne cavity, and then into the rest of the FRIS cavity which is less dense. Beginning in Stage 1, the continental shelf freshens more quickly than the cavity. This creates a density barrier by which the continental shelf (point 3) is less dense than the cavity (point 2) and especially the deep cavity (point 1), impeding the flow of HSSW beneath the ice shelf.

In the Filchner Trough (Fig. 4b), the cavity (point 4) and continental shelf are initially more dense than the offshore WDW (point 5), creating a barrier to onshore flow. In Stage 1, the continental shelf freshens, which allows small incursions of WDW to cross the continental slope. However, the Filchner cavity is still sufficiently dense that the WDW incursions do not enter the cavity; rather, they are overpowered by ISW flowing in the opposite direction. In Stage 2, the cavity has freshened so

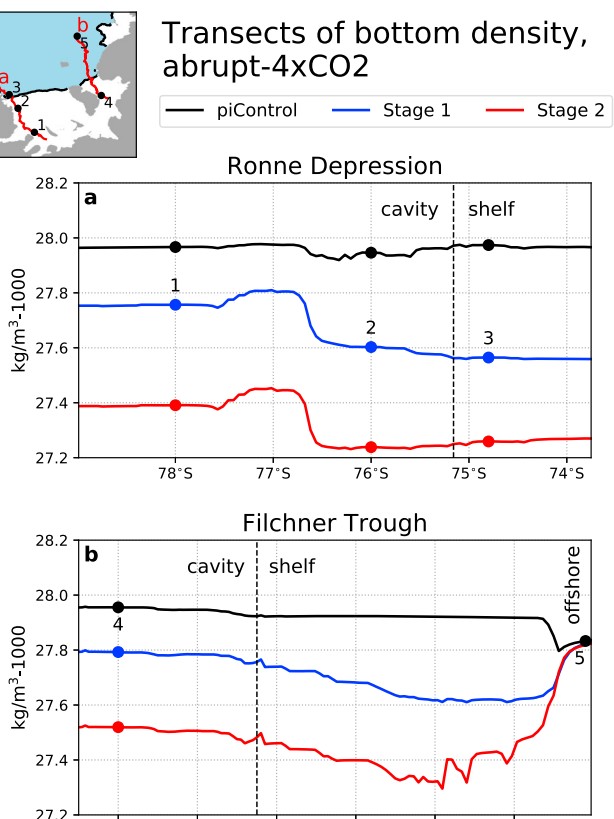

**Fig. 4 Transects of potential density at the bottom of the water column in the FRIS cavity and surrounding continental shelf. a** Ronne Depression; **b** Filchner Trough. The transects are shown in the inset map and consist of the points with the deepest bathymetry at each model latitude, within the given regions. The transects are shown averaged over three time periods: the entire piControl simulation (black), Stage 1 of the abrupt-4xCO2 simulation (years 0–78; blue), and Stage 2 of the abrupt-4xCO2 simulation (years 79–149; red). The points 1–5, as referenced in the text, are marked on both the density plots and the map. The dashed vertical line in each panel shows the location of the ice front. South of this line is the ice shelf cavity, and north of this line is the continental shelf. A small amount of the continental slope and offshore region is shown in the northernmost extent of (**b**).

much that WDW can flow along the entire length of the Filchner Trough and access the ice shelf.

**Stage 1: ice shelf melting decreases**. A reversed density gradient between the inflowing HSSW and the FRIS cavity has profound impacts on the ice shelf. Weaker circulation (Fig. 5a) means that HSSW, the main source of heat to the cavity, is not efficiently renewed. The water within the cavity becomes relatively stagnant, and cools as its heat is extracted by the ice shelf (Fig. 5b). The reduction in thermal driving, particularly near the deep grounding lines, leads to reduced basal melting (Fig. 5c). Compared to the piControl mean, basal mass loss is 20% lower averaged over Stage 1 of 1pctCO2, and 27% lower for abrupt-4xCO2.

Although basal melt rates are lower, the longer residence time induced by slower circulation means that more heat is ultimately extracted from the source HSSW through ice shelf melting. The resulting supercooled ISW, which flows out into Filchner Trough, becomes colder throughout Stage 1 (blue regions in Fig. 6a).

**Stage 2: ice shelf melting increases**. The freshening signal gradually propagates through the FRIS cavity and into the ISW

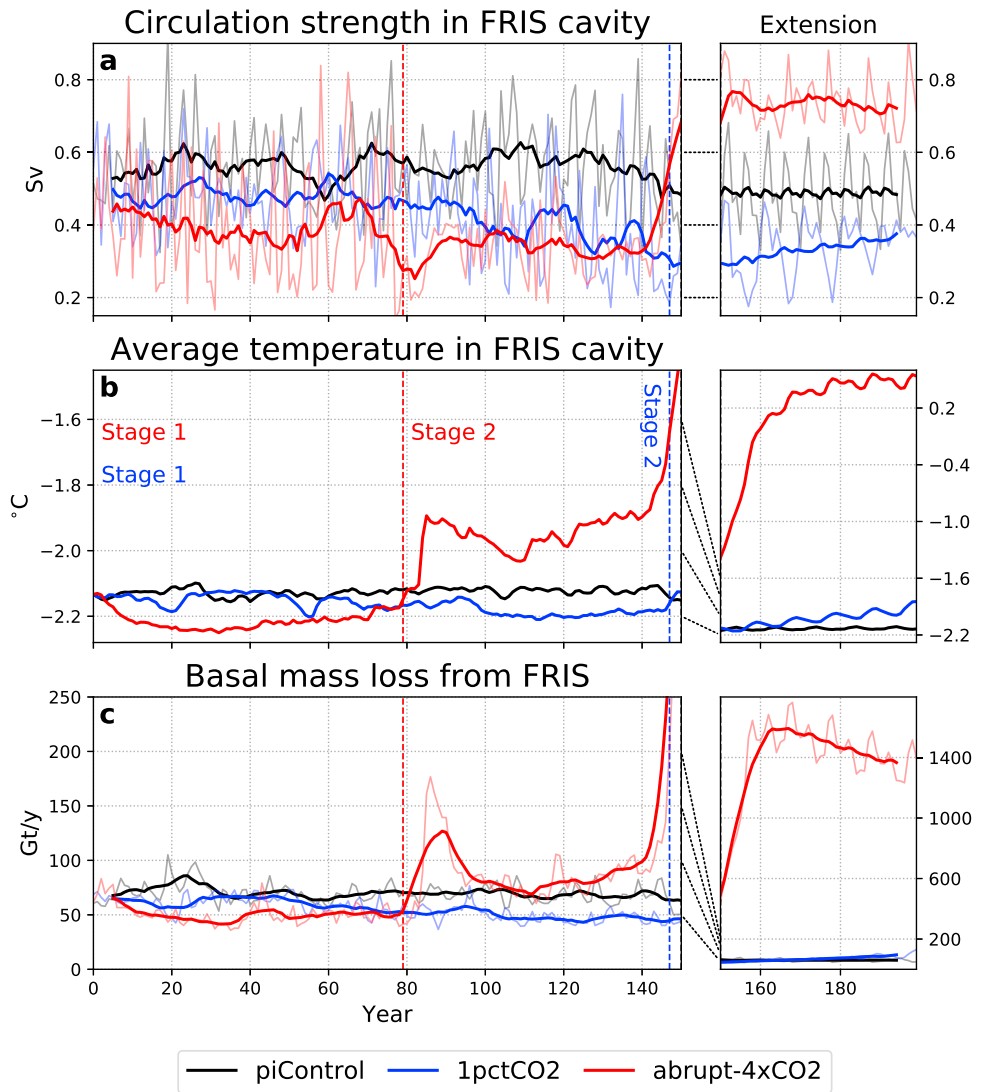

**Fig. 5 Timeseries of simulated properties in the Filchner–Ronne Ice Shelf (FRIS) cavity in both transient scenarios and their 50-year extensions, compared to the pre-industrial control.** Note the different scales between the main simulations (left column) and extensions (right column); the dotted lines joining the columns indicate the relationship between the scales. **a** Circulation strength (Sv), defined by the maximum absolute value of the barotropic streamfunction in the cavity. **b** Temperature ( °C) volume-averaged over the cavity. **c** Basal mass loss (Gt/y) integrated over the ice shelf. Thinner lines in (**a**) and (**c**) show annual averages, while thicker lines show the 11-year running means. In (**b**), only annual averages are shown. Dashed vertical lines show the beginning of Stage 2 in each scenario, as defined in the text. The timeseries, dashed vertical lines, and Stage 1/2 labels are all coloured by simulation, as indicated in the legend.

outflow (Fig. 6b). Eventually, the direction of flow through Filchner Trough reverses, so that pulses of WDW enter the cavity (Fig. 1c). This response is in line with previous analyses[9,11]. Modified WDW also enters the cavity from its original region of inflow, the Ronne Depression (Supplementary Note 3 and Supplementary Movie 1). It follows the existing pathways of HSSW inflow, but the temperature of the water column no longer drops as low as the surface freezing point.

In 1pctCO2, WDW intrusion begins in the last 5 years of the main simulation (model year 145; Fig. 5b). This pulse is repeated each decade of the 50-year extension, indicating that interannual variability in the forcing modulates the initial WDW intrusion. The cavity slowly warms following these regular pulses, and basal mass loss increases (Fig. 5c), but does not yet reach levels substantially higher than pre-industrial.

In abrupt-4xCO2, the first major WDW pulse occurs during years 80–90, causing a warming of ≈0.2 °C in the cavity and a doubling of melt rates compared to piControl (Fig. 5b, c;

Supplementary Movie 1). After this initial pulse, the WDW recedes and melt rates approach mean piControl values, until year 140 when a much larger WDW pulse begins. By the end of the 50-year extension, melt rates are a factor of 21 higher than the piControl mean, and cavity temperatures 2.7 °C warmer. These changes are similar in magnitude to previous studies[7], but the greenhouse forcing is much more extreme.

**Impact on the ice sheet**. The simulated changes in basal melt rates affect ice shelf thickness and the velocity of upstream glaciers. Compared to piControl, during Stage 1 the ice shelf is generally thicker, especially near the deep grounding lines (Fig. 7a). Since weakening circulation beneath FRIS reduces refreezing as well as basal melting, the ice shelf is thinner in regions of net refreezing (central Ronne Ice Shelf and east of Berkner Island; see Fig. 1). The flow of ice streams is generally slower (Fig. 7c), particularly the Slessor Glacier and Foundation

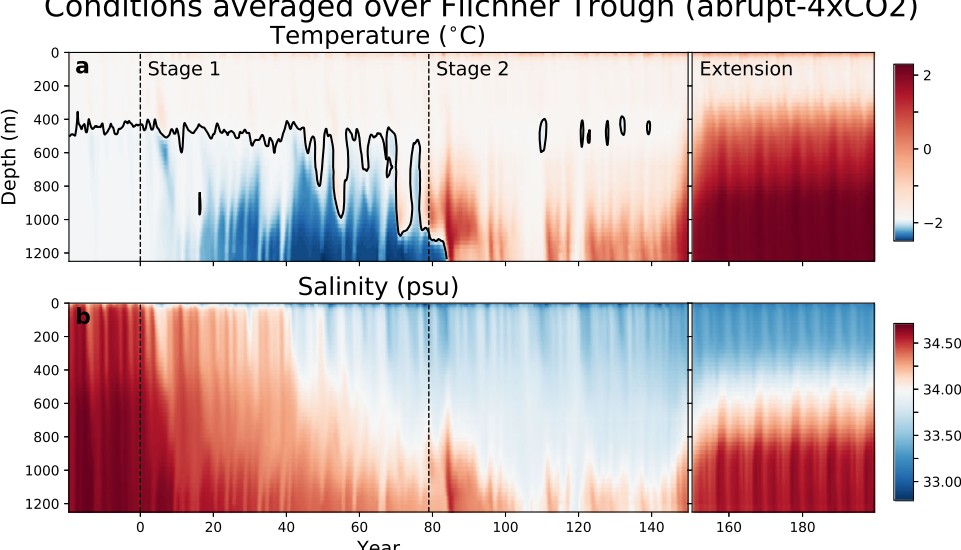

**Fig. 6 Hovmöller plots (depth versus time) averaged over the Filchner Trough region in the abrupt-4xCO2 simulation and the 50-year extension.** To indicate the range of pre-industrial variability, the last 20 years of the spinup are shown prior to year 0. The region is defined in Supplementary Fig. 5. **a** Temperature in °C, with the surface freezing point contoured in black. **b** Salinity in psu. For both variables, the 12-month running mean is shown; note also the nonlinear colour scales. The dashed vertical lines show the beginning of the transient simulation (year 0) and the beginning of Stage 2 (year 79).

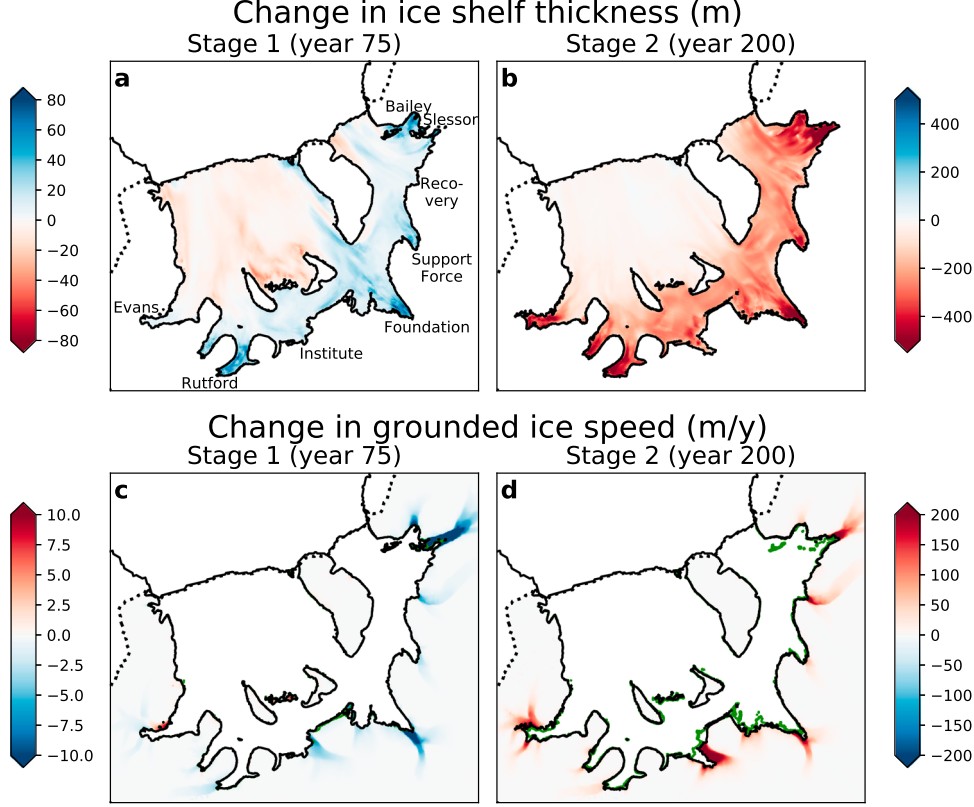

**Fig. 7 Changes in ice sheet properties during the abrupt-4xCO2 simulation and 50-year extension. a** Ice shelf thickness anomaly (m) after 75 years; **b** after 200 years. **c** Grounded ice speed anomaly (m/y) after 75 years; **d** after 200 years. Note the differing colour scales for the two time periods. Model drift is removed by subtracting the piControl simulation at the same point in time. Black grounding lines are from the abrupt-4xCO2 simulation, while the green contours in (**c**) and (**d**) show the piControl grounding lines for comparison. Pinning points in the Filchner Ice Shelf appear as isolated green points. Black labels in (**a**) denote geographical features: the Bailey, Foundation, Institute, Rutford, and Evans Ice Streams; and Slessor, Recovery, and Support Force Glaciers.

Ice Stream, in response to increased buttressing at the grounding line.

These changes are reversed during Stage 2, and by the end of the 50-year extension there is significant and widespread thinning across the ice shelf (Fig. 7b). All of the upstream glaciers accelerate, particularly the Institute Ice Stream and the Slessor Glacier (Fig. 7d). The grounding line retreats relative to piControl, with the most significant retreat near the Institute Ice Stream. The final contribution to global sea level rise is only 1 cm (Methods). However, this rise occurs entirely over the last 40 years of the extension, and is rapidly accelerating. The full response of the ice sheet to increased basal melting of FRIS would occur over timescales much longer than can be simulated with a coupled ice-ocean model.

**Timescales of change**. Stage 1 is a gradual trend, which is initially obscured by decadal variability. It becomes detectable (Methods) after 69 years in 1pctCO2 and 14 years in abrupt-4xCO2, which corresponds to ≈3 °C and ≈5 °C of warming respectively. Observed global mean near-surface air temperature is now ≈1 °C above pre-industrial, which corresponds to ≈30 years of the 1pctCO2 experiment (Fig. 8). Our simulations therefore suggest that Stage 1 is unlikely to be discernible in observations until warming proceeds further.

Stage 2 begins when the Filchner Ice Shelf front transitions from a net exporter of supercooled ISW to a net importer of WDW above the surface freezing point (Methods). This transition occurs after 79 years in abrupt-4xCO2 and 147 years in 1pctCO2. In both cases, this corresponds to ≈7 °C of average global near-surface warming relative to pre-industrial.

UKESM climate projections suggest that 7 °C of warming could only be reached in the 21st century under the highest emissions scenarios. Figure 8 compares global mean near-surface air temperature anomaly in the idealised climate scenarios used in this study (1pctCO2 and abrupt-4xCO2) with more realistic climate change scenarios which could occur over the 21st century (the ScenarioMIP experiments)[18]. It is only the most extreme scenario (SSP5-8.5) which exceeds 7 °C of warming by the end of

the century. The trajectory of SSP3-7.0 suggests it would also exceed this threshold in the 22nd century, but this is not obvious for the cooler scenarios (SSP2-4.5 and SSP1-2.6). With regards to the 3–5 °C warming required for detectability of Stage 1, only the most moderate scenario (SSP1-2.6) stays clearly below this range. In all other scenarios, we would expect Stage 1 to become detectable at some point in the second half of the 21st century.

This comparison should be interpreted with caution, as UKESM is a single model with a higher climate sensitivity than the CMIP6 ensemble mean[22]. The true timescales could therefore be even longer. Other forcing agents in the ScenarioMIP experiments, such as changes in stratospheric ozone, could also have a regional effect on the Antarctic climate[23] which may confound the comparison.

## Discussion

Our simulations are the first to identify a two-timescale response of FRIS melting to a warming climate. This result resolves previous disagreement as to whether FRIS melting would increase or decrease in response to climate change. Stage 1 was previously hypothesised based on one year of observations[6], but no model projections were provided to support this hypothesis. Conversely, previous modelling studies using two closely-related ice-ocean models[7–9] advanced directly to Stage 2 and did not appear to simulate a discernible Stage 1. We hypothesise that these studies may have been overly sensitive to WDW inflow, as they were all forced with the same climate model projection, now two generations old, with no bias corrections. The idealised study of Hazel and Stewart[11] is not comparable to our results, as it applied large step-changes in atmospheric variables which triggered an immediate Stage 2 response, with little opportunity for transient changes.

The challenge now is to bound the timescales of Stage 1 and Stage 2. Our results call for a larger model intercomparison project of ice shelf melting projections in the Weddell Sea, to quantify the model structural uncertainty in these estimates. Internal climate variability may control the precise timing of WDW pulses, so a range of CMIP models and ensemble members

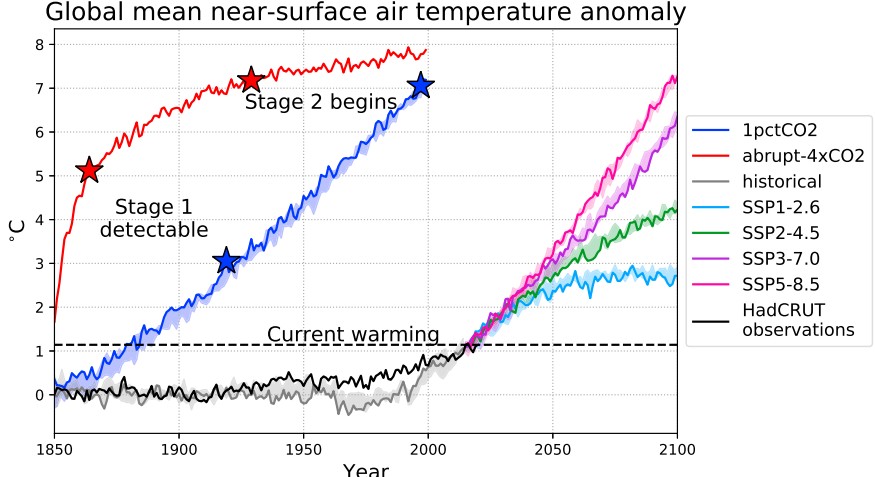

**Fig. 8 Global mean near-surface air temperature anomaly in different CMIP6 scenarios as simulated by UKESM.** Timeseries are annually averaged and shown as anomalies relative to the piControl mean over the last 150 years of simulation. The historical simulation (1850–2014) branches into four SSPs (Shared Socioeconomic Pathways; 2015–2100) as examples of possible future climate change trajectories. The idealised simulations (1pctCO2 and abrupt-4xCO2) are indexed from 1850–1999 for convenience, but do not correspond to any historical forcing. For each scenario, the first ensemble member is shown with a darker line, while the range of four ensemble members is shown with lighter shading (except for abrupt-4xCO2 where only one ensemble member ran the experiment). Stars show the beginning of Stages 1 and 2 as simulated by this study, for the 1pctCO2 and abrupt-4xCO2 cases. Also shown is the observed global mean near-surface air temperature anomaly from the HadCRUT.4.6.0.0 dataset[56], relative to 1850. The dashed line shows the current level of warming, i.e. the HadCRUT observations for 2020.

should be used for forcing, as well as a range of resolutions in the ocean and atmosphere. Fully coupled models, which simulate the atmosphere, ocean, and ice sheet simultaneously, should be included as they are developed. These models would simulate atmosphere-ocean feedbacks, which may be important to FRIS, but which cannot be captured by ice-ocean models such as ÚaMITgcm.

At the same time, observational programmes should monitor the FRIS cavity and continental shelf for signs of Stage 1 beginning. It is not possible to make judgements based on existing observations, which have been sparse until recently. Satellite estimates of basal melt rates[24,25] have not existed for long enough to detect a climatological trend in FRIS melting. The few in-situ measurements which exist beneath the ice shelf show large variability related to the strength of circulation[2,5], which masks any trend in cavity water mass properties. Continued observations would allow for better detection of trends in the region, which could then be used to constrain models. In particular, long-term monitoring should continue in the Filchner Trough[26], as it could detect cooling and freshening of ISW.

Finally, our results suggest that the FRIS region may be less sensitive to climate change than previously thought. In order for melt rates to increase, an extraordinary amount of global warming (7 °C in the UKESM) is required. This far exceeds the pledges of the Paris Agreement, which aims to keep warming below 2 °C. Unless these global mitigation efforts fail dramatically, there is a good chance that the FRIS sector of the Antarctic Ice Sheet could be preserved. However, if global temperatures continue to climb, decreased melting from FRIS should not be considered evidence that the region is unconditionally stable. Our study finds that if continental shelf freshening proceeds far enough, reduced melting will eventually give way to increased melting as warm water floods the cavity. A reduction in FRIS melting could paradoxically be an early warning sign for a later destabilisation of the region.

## Methods
Simulations are performed with a regional Weddell Sea configuration of the ÚaMITgcm model. This coupled framework consists of three parts:

1. the MITgcm ocean model (Massachusetts Institute of Technology general circulation model)[27,28], which includes ice shelf thermodynamics and sea ice
2. the Úa ice flow model[29], which governs the movement and geometry of the ice shelves and ice sheet
3. a custom-built coupler that handles the exchange of information between the two models

**MITgcm ocean.** The ocean component is based on the configuration from Naughten et al.[30], which uses a Weddell Gyre domain with open boundaries at 30°E and 61°S. Horizontal resolution is 0.25 degrees scaled by cosine of latitude, and there are 120 vertical levels with 25 m resolution throughout the FRIS cavity. Bathymetry follows Bedmap2[31] with updates in the Filchner Ice Shelf cavity from Rosier et al.[32] and the grounded iceberg A23-A as defined by Schaffer et al.[33]. The FRIS ice shelf draft is given by Úa as described below; outside the Úa domain (e.g. for the Larsen and Fimbul ice shelves) the ice shelf draft follows Bedmap2. In regions with thin water columns and steeply sloping ice shelves, the bathymetry is deepened by the minimum amount to prevent the formation of spurious subglacial lakes[34], a process hereafter referred to as digging.

To ensure numerical stability on the long timescales used in this study, we do not use the $z^*$ coordinate system as in Naughten et al.[30], but instead employ MITgcm's linear free surface scheme. We use an ice shelf drag coefficient of 0.0025 following new observational evidence[35]. The MITgcm code has also been updated to version 67k. All other parameterisations, including the GM-Redi parameterisation of geostrophic eddies, the KPP vertical mixing scheme, and sea ice elastic-viscous-plastic rheology, follow Naughten et al.[30].

**Úa ice sheet.** Úa is a finite-element ice flow model based on the shallow ice stream approximation, with spatially varying resolution, automatic mesh refinement, and adaptive time-stepping[29,36]. Further numerical details are provided by Gudmundsson[37]. The domain used here consists of the FRIS catchment region[38] with zero inflow conditions on the boundaries. Resolution ranges from 300 m to

50 km, with the finest resolution near grounding lines and ice shear margins. A minimum ice sheet thickness of 10 m is imposed. The ice front is constant in time, and any ice which flows past this boundary is assumed to have calved. The freshwater fluxes resulting from calving are not passed to the ocean, as iceberg freshwater flux is already accounted for in MITgcm (see UKESM Forcing below). This configuration assumes a meteoric ice density of 917 kg/m³ with firn depths provided by the RACMO2.1 firn densification model[39]. The surface mass balance is a climatological record also obtained from RACMO2.1[40] and the basal mass balance is given by MITgcm, as described below. Úa is initialised with the same bathymetry as MITgcm, and ice surface topography from Bedmap2[31]. We invert for the basal slipperiness in the Weertman sliding law and the rate factor in Glen's Flow Law, using surface ice velocity from the MEaSUREs 2 dataset[41] which covers the years 1996–2016. The inversion is based on an iterative adjoint method with Thikonov regularisation, and minimises the cost function between observations and model output of surface velocities. There is an additional term in the cost function to penalise large changes in ice thickness, which reduces drift in the subsequent simulations.

**Coupling.** The coupling algorithm originated with De Rydt and Gudmundsson[42] but has been rewritten and extended so it is suitable for a realistic ocean with time-varying forcing and sea ice. For each coupling segment, Úa and MITgcm run independently for 1 year, then stop, exchange data, and restart. This design is known as offline coupling, and is standard practice for most ice sheet-ocean models in the Marine Ice Sheet Ocean Model Intercomparison Project (MISOMIP)[43].

At each coupling step, MITgcm sends Úa the ice shelf basal melt rates, which Úa then linearly interpolates to all ice shelf nodes. No basal melting is applied in partially grounded elements, to avoid unphysical melting at connected nodes upstream of the grounding line. Úa sends MITgcm the ice shelf geometry and floating mask, linearly interpolated to the MITgcm grid. Each MITgcm cell containing at least one Úa ice shelf node is considered to be floating.

The MITgcm restart file is then edited to suit the new geometry. As much information as possible is preserved from the previous ocean state, unlike the re-initialisation approach of De Rydt and Gudmundsson[42]. Newly opened cells are filled with the properties (temperature, salinity, and free surface) of nearest neighbours, preferencing vertical neighbours. This ensures that newly opened cells at the ice base are filled with conditions from the ice-ocean boundary layer wherever possible. Newly closed cells become part of the land mask with no changes to the properties of surrounding cells.

The velocity in newly opened cells is first set to zero. If a given water column is newly opened, its velocity remains at zero. Otherwise, the velocity is corrected to preserve the barotropic transport field, by adding or subtracting a small value from the entire water column. This correction is necessary to ensure numerical stability. The barotropic transport field is not preserved in the case where an entire water column closes.

The pressure load anomaly of the ice shelf is recalculated at each coupling step, assuming the water displaced by the ice shelf has the temperature and salinity of the uppermost ocean layer. Digging of the MITgcm bathymetry is recalculated each coupling step as the ice shelf draft changes, and previous digging is reversed if it is no longer necessary. Úa does not see these changes in the bathymetry.

**UKESM forcing.** MITgcm is forced with output from the UKESM1-0-LL submission to CMIP6[44–46]. This forcing consists of daily fields for atmospheric variables (near-surface temperature, specific humidity, and winds; surface downwelling shortwave and longwave radiation; sea level pressure; and precipitation) and monthly fields for ocean and sea ice variables on the lateral boundaries (ocean temperature, salinity, and horizontal velocity; sea ice area, thickness, velocity, and snow thickness). The atmospheric fields are used to calculate bulk fluxes within the MITgcm code, which then force the ocean and sea-ice models. We use the first ensemble member of UKESM (r1i1p1f2).

While this version of UKESM does not include ice shelf cavities, it prescribes a steady three-dimensional coastal freshwater flux representing ice shelf basal melting[47,48]. The spatial distribution of this flux follows the observations of Rignot et al.[24]. UKESM also considers steady iceberg calving through a statistical iceberg model[49]. To conserve mass, the sum of the meltwater flux and the calving flux is equal to the modelled pre-industrial surface mass balance of Antarctica (not including surface runoff, which is a separate flux to the ocean). This freshwater flux implicitly enters the MITgcm domain via the lateral boundary conditions. Therefore, the meltwater from ice shelves and icebergs is accounted for both within the ÚaMITgcm domain and in the external forcing from UKESM.

The initial conditions in MITgcm are a combination of UKESM output for the first month of simulation, and the Southern Ocean State Estimate (SOSE)[50] January climatology. SOSE conditions for temperature and salinity are applied on the continental shelf (defined by the 2500 m isobath on the continental slope) and extrapolated into ice shelf cavities. UKESM temperature and salinity are used elsewhere in the domain, and sea ice initial conditions (sea ice area, thickness, and snow thickness) come exclusively from UKESM. This merged approach to initial conditions was chosen due to a poor representation of the continental shelf in the low-resolution UKESM ocean.

The MITgcm configuration has been shown to produce realistic results when forced with atmospheric reanalyses[30]. In this domain, the only significant bias of UKESM's historical simulation compared to the ERA5 reanalysis[51] is the coastal

winds around Antarctica. These winds are generally too weak and diffuse, which is expected given the low resolution of UKESM. Coastal winds, including katabatic winds and barrier winds, are directed by small-scale features of the ice sheet surface topography which can only be properly captured at high atmospheric resolution[52]. We counteract this bias using a coastal wind correction, which is applied to the UKESM output when it is passed to MITgcm. This method extends the work of Mathiot et al.[52] by applying a spatially varying, time-constant correction to winds near the coastline. The winds are converted to local polar coordinates, and are scaled and rotated such that their time-averaged magnitude and angle agrees with the ERA5 reanalysis over the observational period. Further details of the correction are provided in Supplementary Note 1. With this correction, present-day conditions simulated by ÚaMITgcm in the FRIS region largely agree with observations, as shown in Supplementary Note 2.

To prevent sea level drift in MITgcm, the ocean velocity boundary conditions are modified each year. This is necessary due to differences in the UKESM and MITgcm freshwater budgets which would otherwise lead to a net convergence or divergence of volume within the domain. The correction adds or subtracts a small value from the normal velocities at the open boundaries, to produce a volume flux, which relaxes the area-averaged free surface to zero over a 1-year timescale. Interannual variability in the freshwater budget means that this value needs to be updated every year, as a rolling correction.

Freshwater fluxes from iceberg melt are given by the monthly climatology of Storkey et al.[53] and do not change over the simulations. Similarly, the surface mass balance used to force Úa remains constant in time.

**Spinup**. ÚaMITgcm was spun up in three stages to ensure numerical stability and consistency between the two component models:

1. An initial standalone ice sheet simulation of 10 years was performed with Úa, forced with basal melt rates from the piControl average of a standalone MITgcm simulation performed previously. The purpose of this Úa simulation was to overcome the initial adjustment period, which can trigger brief but rapid changes in ice geometry. The final geometry from this simulation was used to redefine the MITgcm domain.
2. An ocean-only spinup of 10 years was then performed, forced by years 2880–2889 of the piControl simulation in UKESM. These years were chosen so that the transient simulations would later branch from the correct point of piControl.
3. The coupled ice sheet was activated for a further 20 years of spinup under piControl forcing (years 2890–2909).

Three simulations branched from this point, each running for 150 years: the continuation of the piControl case (years 2910–3059), abrupt-4xCO2, and 1pctCO2. The final decade of each forcing scenario was then repeated 5 times for a 50-year extension.

**Calculation of Stage 1 and Stage 2 thresholds**. To determine the threshold of detectability for Stage 1, we compare two annually averaged timeseries (FRIS cavity temperature and basal mass loss, as in Fig. 5b, c) between each transient simulation and piControl. For each variable $v$ and each year $t$, we calculate the mean $\mu$ and standard deviation $\sigma$ of the 20-year period around $t$. The variable $v$ is deemed to be outside the bounds of pre-industrial variability when the range $(\mu-\sigma, \mu+\sigma)$ no longer overlaps between the two simulations. More specifically, when

$$\mu(v_{(t-10:t+9)}) + \sigma(v_{(t-10:t+9)}) < \mu_{PI}(v_{(t-10:t+9)}) - \sigma_{PI}(v_{(t-10:t+9)}) \qquad (1)$$

where $\mu$ and $\sigma$ refer to the transient simulation (abrupt-4xCO2 or 1pctCO2), and $\mu_{PI}$ and $\sigma_{PI}$ to piControl. Stage 1 is considered to be detectable for the first value of $t$ where this holds for both variables, i.e. cooler temperature and lower basal mass loss.

To determine the threshold at which Stage 2 begins, we calculate the temperature difference from the surface freezing point ($T-T_f$), which varies with the local salinity but not with depth. This quantity is averaged over the ocean cells at the Filchner Ice Shelf front at all depths between the ice shelf base and the seafloor. If the integrated result is negative, this transect has net supercooling with respect to the surface freezing point, indicating a dominance of Ice Shelf Water. This is the case for every year of piControl. Stage 2 begins when the annually averaged value first becomes positive, indicating a new regime of warm water transport into the ice shelf cavity.

**Calculation of sea level rise contribution**. We follow the method of Goelzer et al.[54] to convert changes in total ice volume above flotation (VAF) to a global sea level contribution (see their Eq. 2). To account for drift in the ice sheet model, we use the difference in VAF between the transient simulation (here abrupt-4xCO2) and the piControl simulation at the same point in time.

## Data availability
The UKESM output used to force the model is freely available in the CMIP6 archive[44–46]. The output from the ÚaMITgcm simulations can be obtained from the corresponding author upon request.

## Code availability
The latest version of the ÚaMITgcm source code is publicly accessible at https://github.com/knaughten/UaMITgcm. These simulations use the WSFRIS configuration, and the specific version of ÚaMITgcm used here has been archived[55] with doi:10.5281/zenodo.3876453. All pre- and post-processing code are publicly accessible at https://github.com/knaughten/mitgcm_python.

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

## Acknowledgements

The authors are grateful to Hilmar Gudmundsson for developing and maintaining the Úa source code, and to Keith Nicholls, Keith Makinson, and John King for helpful discussions while writing this manuscript. David Munday provided advice regarding the MITgcm salt budget analysis, and Markus Janout provided the PS111 observational data for model evaluation. Computational resources were provided by the ARCHER UK National Supercomputing Service as well as the JASMIN data analysis facility. This research is part of the Filchner Ice Shelf System (FISS) project NE/L013770/1.

## Author contributions

K.A.N. devised and carried out the experiments, analysed the output, and wrote the manuscript. K.A.N., J.D.R., and S.H.R.R. developed the ÚaMITgcm model. A.J. and P.R. H. assisted with experimental design and model troubleshooting. J.K.R. provided the UKESM forcing and additional analysis. All authors contributed to the development of ideas and provided feedback on the manuscript.

## Competing interests

The authors declare no competing interests.
