## [Peer Review File · Nature Communications]

REVIEWER COMMENTS

Reviewer #1 (Remarks to the Author):

Review: 'Two-timescale response of a large Antarctic ice shelf to climate change' by Naughten et al.

In this study the authors investigate possible future changes in the melt rate of the Filchner-Ronne Ice Shelf (FRIS) using a coupled ocean/sea ice/ice shelf regional model. Some previous studies have indicated that global warming may reduce the melt of the FRIS by suppressing the formation of the High Salinity Shelf Waters (HSSW) that are observed to circulate beneath it, while others have indicated that the FRIS might undergo an abrupt transition to a rapidly-melting state, driven by inflow of Warm Deep Water (WDW) from offshore. To reconcile these previous lines of evidence the authors force their regional model with bounding oceanic and atmospheric states drawn from three different earth system model projections: a preindustrial control and two extreme increased-CO₂ scenarios. They show that warming initially leads to a reduction in FRIS melt, lasting several decades (Stage 1), before a transition to substantially increased melt due to WDW intruding beneath the FRIS (Stage 2). They use a freshwater budget for the southern Weddell Sea continental shelf to deduce that these transitions are primarily driven by changes in surface freshwater fluxes - primarily an increase in sea ice melting closer to the Antarctic continent. However, the authors find that the transition to increased FRIS melt requires an extreme warming of around 7C to occur, suggesting that this transition is unlikely to occur during the 21st Century. Their results indicate that we are more likely to see reductions in FRIS melt during the 21st Century, preceding a later transition to increased melt if global warming continues unabated.

This study addresses a significant topic in the field of climate science, and resolves what previously appeared to be conflicting lines of evidence regarding the future of the FRIS. The manuscript was a pleasure to read, with very clear text and figures. My only presentational complaint is that I had to continuously switch between the main text, the Methods, and the Supplementary Information, but I suspect that this was imposed on the authors by Nature's inexplicable article structure/length requirements. The description of the methodology is particularly detailed, which was most welcome.

Below I have included some comments for the authors' consideration. They are generally minor, and remarkably few (for this reviewer), reflecting the high quality of this study. I therefore recommend acceptance subject to minor revision.

Comments/questions:

1. The authors note that the transition of the FRIS to a high-melt state has been identified in a few previous studies (Hellmer et al. 2012, Hellmer et al. 2017, Hazel & Stewart 2020). I read these studies as part of my review, and noted two key differences from the authors' diagnostics: (i) the "Stage 1" reduction in FRIS melt is not (visually) evident in the time series presented in these previous studies, and (ii) the transition to rapid FRIS melt seems to occur much more rapidly (within a couple of decades) in these previous studies. It would be appropriate to discuss these differences (if indeed there are differences) in the manuscript, and offer some kind of explanation - is there some process that is captured by the authors' model that was not captured in previous studies?
2. The atmospheric forcing and open ocean boundaries are forced using output from the low-resolution UKESM. The authors therefore correct the wind forcing because the UKESM cannot resolve the coastal winds. However, they do not appear to have corrected the oceanic boundary conditions. I think this may be a more significant caveat than the authors have indicated: high-resolution models (e.g. 1/10

degree horizontal grid spacing) simulate substantially different coastal dynamics from low-resolution ESMs (Newsom et al. 2016, J. Climate; Dufour et al. 2017, J. Climate), and in particular the coastal freshwater dynamics may change qualitatively if simulated at sufficiently high resolution (Moorman et al. 2020, J. Climate). In particular, Moorman et al. show that increased ice shelf melt (e.g. due to global warming) leads to strengthening of the Antarctic Slope Front and isolation of the continental shelf from the open ocean. Though the authors find that surface freshwater fluxes are the primary driver of changes in FRIS melt in their simulations, with a relatively minor contribution due to advection from upstream, this conclusion could partly be an artifact of the low-resolution UKESM upstream boundary conditions. If upstream influences were accurately represented then this could conceivably change the driver of changes in FRIS melt, and modify the authors' conclusions about the 7C warming threshold. I think this warrants further discussion in the manuscript (and future study!).

3. The authors associate the changes in the FRIS state (Stages 1 and 2) to changes in the density difference between the Filchner Trough and offshore, and between the Ronne depression and the cavity. While these metrics are intuitive and in line with previous studies, I have a few concerns: First, the authors use density differences between two specific points in 3D space to define their metrics, which I would expect to make the metrics rather sensitive - why not some suitable integrated measures?

Second, it seems that the density difference between the offshore waters and the Filchner Trough does not accurately predict the transition from Stage 1 to Stage 2 - the transition does not occur when the density difference passes zero. Have the authors considered alternative metrics, e.g. the density difference between the offshore waters and the FRIS cavity?

Finally, I think the authors overstate the case for these density differences "modulating" the FRIS response. Yes, there is visibly a relationship between changes in these density differences and the Stage of the FRIS, but the authors have not demonstrated a causal relationship. I suggest the authors soften their conclusions on this particular point.

4. In Figure 2, the authors use the area-averaged absolute magnitude of the barotropic streamfunction to quantify the cavity circulation strength. However, this is a difficult metric to interpret - what does this actually quantify? I suspect it's not misleading, but I encourage the authors to consider whether a more intuitive/relatable metric could be used here instead.

Reviewer #2 (Remarks to the Author):

My full review is attached as a PDF.

I wish my name to be relayed to the authors, as I do not support the practice of anonymous review.

Xylar Asay-Davis

Reviewer #3 (Remarks to the Author):

Review of Naughten, De Rydt, Rosier, Jenkins, Holland, and Ridley: "Two-timescale response of a large Antarctic ice shelf to climate change" (Nature Communication, Paper: NCOMMS-20-39498-T)

The manuscript of Naughten and others simulate with a regional coupled ocean-ice sheet model system the response of the Filchner-Ronne-Ice Shelf (FRIS) under warming climates scenarios. Since FRIS is sustained by ice streams fed by large drainage basins of the Eastern and Western Antarctic Ice Sheets, a disintegration of FRIS would lead to sustained ice loss of these drainage basins. This loss would potentially raise the global sea level substantially.

The model system comprises of the ocean model MITgcm and ice sheet model Úa. The system is

driven by corrected forcing extracted from the global earth system model UKESM for three climate scenarios: pi-industrial climate (piControl), and two idealized future scenarios where four-times the pre-industrial atmospheric conditions is reached immediately (abrupt4xCO2) or gradually with a 1% rise per year (1pctCO2). The model shows two distinct responses, which have been described in the literature before as opponent responses. The authors show convincingly that the formerly contradicting reactions are instead interrelated. They argue that former studies may have only reached the so-called "Stage 1" or have advanced directly into the "Stage 2".

Intermediate warming reduces sea ice formation and the production of High Salinity Shelf Water, which lowers the density gradient between the FRIS cavern and the ambient ocean. This gradient drives the inflow predominately on the western side. Therefore, it reduces the ventilation of the FRIS cavern. Consequently, the mean temperature in the cavern cools, and the integrated basal melting decreases by 20% to 27%. This configuration is labeled as Stage 1. Over time, the temperature drop in FRIS also influences the density gradient on FRIS' eastern side. This is the link between both stages. As soon as this second density gradient drops below a probably model-dependent threshold, the ocean circulation reverses the flow direction. The outflow of cold Ice Shelf Water (ISW) is replaced by Warm Deep Water (WDW) entering the cavern. Consequently, the average temperature and the related basal melting rates skyrocket in FRIS. Ultimately, the grounding line retreats, and ice streams feeding the Filchner-Ronne Ice Shelf accelerates, which raises the sea level.

This study is highly relevant for several reasons.

1. It solves a before not understood difference between independent model simulations.
2. An apparent drop of the basal melting rate is commonly understood as an indication of a stable state. However, this study highlights that it could be a harbinger of a situation where the ice shelf's disintegration becomes more likely or even inevitable.
3. A more nuanced understanding and observation strategy are probably necessary to monitor the stability of some ice shelves surrounding Antarctica.

The study is well organized and written. The figures are of high quality, necessary, and informative. It was a pleasure to read this excellent manuscript.

I recommend the publication of the manuscript after a few minor corrections.

General comments

=====

Since changing ocean conditions do not feedback to the atmosphere, which provides the upper boundary conditions for the coupled ocean-ice sheet simulation, the authors have probably used so-called mixed boundary conditions for their setup (Mikolajewicz and Maier-Reimer 1994). These conditions are known to produce artifacts for, at least, the Atlantic Meridional Overturning Circulation (AMOC). Though restricted to a limited region, are your simulation also subject to artifacts because the feedbacks between atmosphere and ocean are not represented?

Since the UKESM is used as a source for the forcing that drives ÚaMITgcm, it is unclear how UKESM's biases impact the presented conclusion. More specifically: How does the reported temperature bias of 0.5-2°C in the Southern Ocean (Sellar et al. 2019) affect your results? Since it is still common in global climate and earth system models that open-ocean deep convection ventilates the deep Weddell Sea via the rarely occurring Weddell Sea Polynia, is UKESM also subject to this bias? Since deep convection at the wrong place has a distinct influence on basal melting in FRIS (Naughten et al. 2019). If the UKESM shows the deep ocean ventilation bias, how do the related fluxes that probably drive the regional model ÚaMITgcm, imprint its bias on the presented results?

Specific comments

=====

Main document

Page 4, Line 66: I guess your mean: "global climate models have reduced their atmospheric biases."

Page 11, Line 154: You may extent to: "in 1pctCO4, WDW intrusion begins in the last 5 years (model year 145, 5 years after four-times CO2 has been reached) of the main simulation."

Page 11, Line 156-157: I do not understand immediatly why "the WSW intrusion are imprinted in the atmospheric forcing." Please clarify this point.

Page 12, Line 173: I guess you talk about the near-surface or 2m air-temperature and not the actual surface temperature. Could you please clarify?

Page 13, Line 181: Here, I guess you mean near-surface and not surface temperature.

Page 13, Line 191-192: You say "(central Ronne Ice Shelf and east of Berkner Island)." Please refer the reader to Figure 1 or add the label to Figure 4 for Berkner Island.

Page 16, Line 233-234: The extraordinary warming of 7°C might be characteristic of the chosen global climate model UKESM. Hence, you may rewrite it to "amount of global warming (e.g., 7°C in UKESM)."

Page 19, Line 281: I refer to the process named "digging," which may lead to an overly sensitive response. What would happen if you would seal/fill spurious lakes, and how would it influence your results?

Page 21, Line: 325-326: Would the sensitivity increase if you would allow basal melting in partially floating elements proportional to the floating fraction?

Figure 2: You may add to avoid confusion about colored labels: "The color of the time-series, vertical dashed lines, and labels correspond to the scenarios listed in the legend at the bottom."

Figure 4: Is it correct that the green line representing the grounding line in piControl appears to be a broken line?

Supplement Material

Page 15, Line 195: Do you mean near-surface air temperature?

Bibliography

Response to Reviewers

This document is colour-coded as follows:

- Comments by reviewers are in **blue**.
- Our responses are in **black**.
- Blocks of text we have added to the manuscript are in **red**.

Reviewer 1

In this study the authors investigate possible future changes in the melt rate of the Filchner-Ronne Ice Shelf (FRIS) using a coupled ocean/sea ice/ice shelf regional model. Some previous studies have indicated that global warming may reduce the melt of the FRIS by suppressing the formation of the High Salinity Shelf Waters (HSSW) that are observed to circulate beneath it, while others have indicated that the FRIS might undergo an abrupt transition to a rapidly-melting state, driven by inflow of Warm Deep Water (WDW) from offshore. To reconcile these previous lines of evidence the authors force their regional model with bounding oceanic and atmospheric states drawn from three different earth system model projections: a preindustrial control and two extreme increased-CO₂ scenarios. They show that warming initially leads to a reduction in FRIS melt, lasting several decades (Stage 1), before a transition to substantially increased melt due to WDW intruding beneath the FRIS (Stage 2). They use a freshwater budget for the southern Weddell Sea continental shelf to deduce that these transitions are primarily driven by changes in surface freshwater fluxes - primarily an increase in sea ice melting closer to the Antarctic continent. However, the authors find that the transition to increased FRIS melt requires an extreme warming of around 7C to occur, suggesting that this transition is unlikely to occur during the 21st Century. Their results indicate that we are more likely to see reductions in FRIS melt during the 21st Century, preceding a later transition to increased melt if global warming continues unabated.

This study addresses a significant topic in the field of climate science, and resolves what previously appeared to be conflicting lines of evidence regarding the future of the FRIS. The manuscript was a pleasure to read, with very clear text and figures. My only presentational complaint is that I had to continuously switch between the main text, the Methods, and the Supplementary Information, but I suspect that this was imposed on the authors by Nature's inexplicable article structure/length requirements. The description of the methodology is particularly detailed, which was most welcome.

We are glad that the reviewer enjoyed reading our paper and found it to be a significant contribution to the field.

We agree the reliance on supplementary material is suboptimal. This manuscript was transferred from another Nature journal with stricter limits on article length. Luckily, the editor has informed us that Nature Communications has more generous allowances, so we have now merged key sections of the supplementary material into the main text:

- The previous Supplementary Section 3 ("Causes of freshening") is now merged into the main text section "Freshening of continental shelf"

- The previous Supplementary Section 4 (“Density gradients”) is now in the main text as a new section “Changes in density gradients”, and has been largely rewritten as detailed later in our response
- The previous Supplementary Section 6 (“Comparison of climate scenarios”) is now merged into the main text section “Timescales of change”

The Methods section remains separate at the end of the main text, as per Nature requirements.

Below I have included some comments for the authors’ consideration. They are generally minor, and remarkably few (for this reviewer), reflecting the high quality of this study. I therefore recommend acceptance subject to minor revision.

Comments/questions:

1. The authors note that the transition of the FRIS to a high-melt state has been identified in a few previous studies (Hellmer et al. 2012, Hellmer et al. 2017, Hazel & Stewart 2020). I read these studies as part of my review, and noted two key differences from the authors’ diagnostics: (i) the “Stage 1” reduction in FRIS melt is not (visually) evident in the time series presented in these previous studies, and (ii) the transition to rapid FRIS melt seems to occur much more rapidly (within a couple of decades) in these previous studies. It would be appropriate to discuss these differences (if indeed there are differences) in the manuscript, and offer some kind of explanation - is there some process that is captured by the authors’ model that was not captured in previous studies?

We have expanded our discussion of previous studies in the “Implications” section (lines 288-302):

Our simulations are the first to identify a two-timescale response of FRIS melting to a warming climate. This result resolves previous disagreement as to whether FRIS melting would increase or decrease in response to climate change. Stage 1 was previously hypothesised based on one year of observations (Nicholls, 1997), but no model projections were provided to support this hypothesis. Conversely, previous modelling studies using two closely-related ice-ocean models (Hellmer et al., 2012; Timmermann and Hellmer, 2013; Hellmer et al., 2017) advanced directly to Stage 2 and did not appear to simulate a discernible Stage 1. We hypothesise that these studies may have been overly sensitive to WDW inflow, as they were all forced with the same climate model projection, now two generations old, with no bias corrections. The idealised study of Hazel and Stewart (2020) is not comparable to our results, as it applied large step-changes in atmospheric variables which triggered an immediate Stage 2 response, with little opportunity for transient changes.

2. The atmospheric forcing and open ocean boundaries are forced using output from the low-resolution UKESM. The authors therefore correct the wind forcing because the UKESM cannot resolve the coastal winds. However, they do not appear to have corrected the oceanic boundary conditions. I think this may be a more significant caveat than the authors have indicated: high-resolution models (e.g. 1/10 degree horizontal grid spacing) simulate substantially different coastal dynamics from low-resolution ESMs (Newsom et al. 2016, J. Climate; Dufour et al. 2017, J. Climate), and in particular the coastal freshwater dynamics may change qualitatively if simulated at sufficiently high resolution (Moorman et al. 2020, J.

Climate). In particular, Moorman et al. show that increased ice shelf melt (e.g. due to global warming) leads to strengthening of the Antarctic Slope Front and isolation of the continental shelf from the open ocean. Though the authors find that surface freshwater fluxes are the primary driver of changes in FRIS melt in their simulations, with a relatively minor contribution due to advection from upstream, this conclusion could partly be an artifact of the low-resolution UKESM upstream boundary conditions. If upstream influences were accurately represented then this could conceivably change the driver of changes in FRIS melt, and modify the authors' conclusions about the 7C warming threshold. I think this warrants further discussion in the manuscript (and future study!).

In our model configuration, the open boundaries (30°E and 61°S) are relatively far-field from the main region of study (FRIS). In particular, the eastern boundary is far enough upstream that a sizable portion of the coastal current is simulated within the MITgcm domain, where the relatively high resolution allows for a better representation of coastal dynamics than in UKESM. Furthermore, as shown in our response to Reviewer 3, UKESM's water mass properties at the lateral boundaries agree well with observations, so it was not necessary to apply bias corrections here. For these reasons, we do not expect that the boundary conditions introduce first-order uncertainty into our results. However, we have added a brief discussion of forcing resolution into the text (lines 145-148):

Note that the relatively low resolution of UKESM may decrease confidence in its ability to accurately simulate coastal dynamics (Newsom et al. 2016). Therefore, some uncertainty exists in the relative role of advection in this salt budget, which could be important for FRIS melt rates (Bull et al., 2020).

We have also mentioned model resolution as an important factor in forcing our planned future intercomparison (lines 306-309):

Internal climate variability may control the precise timing of WDW pulses, so a range of CMIP models and ensemble members should be used for forcing, as well as a range of resolutions in the ocean and atmosphere.

3. The authors associate the changes in the FRIS state (Stages 1 and 2) to changes in the density difference between the Filchner Trough and offshore, and between the Ronne depression and the cavity. While these metrics are intuitive and in line with previous studies, I have a few concerns:

First, the authors use density differences between two specific points in 3D space to define their metrics, which I would expect to make the metrics rather sensitive - why not some suitable integrated measures?

Second, it seems that the density difference between the offshore waters and the Filchner Trough does not accurately predict the transition from Stage 1 to Stage 2 - the transition does not occur when the density difference passes zero. Have the authors considered alternative metrics, e.g. the density difference between the offshore waters and the FRIS cavity?

Finally, I think the authors overstate the case for these density differences "modulating" the FRIS response. Yes, there is visibly a relationship between changes in these density differences and the Stage of the FRIS, but the authors have not demonstrated a causal relationship. I suggest the authors soften their conclusions on this particular point.

We have rewritten the section on density changes, and have replaced the original analysis (of density differences between points) with density transects along the Ronne Depression and Filchner Trough at different time periods within the simulation. The new analysis gives a more complete picture of density changes along the two flow pathways. We have also noted that while density is an important driver of circulation, the system is too complex for the direction of flow to be solely determined by a simple metric.

This section, which has now been merged into the main text, reads as follows (lines 163-196):

Continental shelf freshening has implications for the density gradients which drive circulation beneath FRIS. There are two density gradients to consider:

1. between the dense source HSSW (in Ronne Depression and over Berkner Bank) and the less dense FRIS cavity; and
2. between the dense outflowing ISW (in Filchner Trough) and the less dense offshore WDW.

A reversal of the first density gradient characterises Stage 1 of the response, when HSSW inflow slows and ice shelf melting decreases. A reversal of the second density gradient characterises Stage 2, when WDW flows into the cavity and ice shelf melting increases.

Figure 4 illustrates how these density gradients change as the climate warms and the continental shelf freshens. The complexity of ocean circulation means that individual density transects are not simple determinants of the direction of flow, which also depends on topography, wind stress, and many other factors. However, changes in density gradients do influence, and can reasonably explain, circulation changes seen in the simulation (see Supplementary Video 1).

In the Ronne Depression (Figure 4a), bottom density is initially relatively constant along the entire length of the depression. HSSW flows freely downslope, southward from the continental shelf into the western Ronne cavity, and then into the rest of the FRIS cavity which is less dense. Beginning in Stage 1, the continental shelf freshens more quickly than the cavity. This creates a density barrier by which the continental shelf (point 3) is less dense than the cavity (point 2) and especially the deep cavity (point 1), impeding the flow of HSSW beneath the ice shelf.

In the Filchner Trough (Figure 4b), the cavity (point 4) and continental shelf are initially more dense than the offshore WDW (point 5), creating a barrier to onshore flow. In Stage 1, the continental shelf freshens, which allows small incursions of WDW to cross the continental slope. However, the Filchner cavity is still sufficiently dense that the WDW incursions do not enter the cavity; rather, they are overpowered by ISW flowing in the opposite direction. In Stage 2, the cavity has freshened so much that WDW can flow along the entire length of the Filchner Trough and access the ice shelf.

Figure 4: Transects of potential density at the bottom of the water column along (a) Ronne Depression and (b) Filchner Trough. The transects are shown in the inset map and consist of the points with the deepest bathymetry at each model latitude, within the given regions. The transects are shown averaged over three time periods: the entire piControl simulation (black), Stage 1 of the abrupt-4xCO₂ simulation (years 0-78; blue), and Stage 2 of the abrupt-4xCO₂ simulation (years 79-149; red). The points 1-5, as referenced in the text, are marked on both the density plots and the map. The dashed vertical line in each panel shows the location of the ice front. South of this line is the ice shelf cavity, and north of this line is the continental shelf. A small amount of the continental slope and offshore region is shown in the northernmost extent of (b).

4. In Figure 2, the authors use the area-averaged absolute magnitude of the barotropic streamfunction to quantify the cavity circulation strength. However, this is a difficult metric to interpret - what does this actually quantify? I suspect it's not misleading, but I encourage the authors to consider whether a more intuitive/relatable metric could be used here instead.

As suggested by Reviewer 2, we have changed this timeseries to the maximum absolute value of the barotropic streamfunction. This metric is more intuitive, and shows the same pattern of weakening circulation in Stage 1 which later strengthens during Stage 2. More details can be found in our response to Reviewer 2's comment.

We would like to thank Reviewer 1 for their time in reading and reviewing our manuscript, and for their suggestions which helped to strengthen and clarify our analysis.

Reviewer 2 (self-identified as Dr Xylar Asay-Davis)

General Comments:

This paper describes three simulations used to explore the effects of climate change on melting below the Filchner-Ronne Ice Shelf (FRIS). The authors find that melting evolves in two stages, first a reduction due to freshening on the continental shelf that reduces circulation in the FRIS cavity, then a significant increase over a few decades due to a significant increase in available heat when warmer modified Weddell Deep Water reaches the FRIS cavity. The simulations suggest that a mean global warming of about 7°C is required before this second stage of melt can take place, suggesting that it is unlikely to occur in the next century except under the highest emissions scenarios. These results are very compelling, and an interesting contrast to earlier findings such as Hellmer et al. (2012), showing increased FRIS melting with less dramatic amounts of global warming. The authors suggest that significant biases in the Earth System Model (ESM) output in these previous studies may have biased these simulations to be too susceptible to increased FRIS melting. Nevertheless, in the Implications section and section 6 of the supplement, the authors rightly point out that their findings are from only one coupled ice sheet-ocean (UaMITgcm) used to downscale three idealized scenarios from a single ESM (UKESM), and that simulations with additional models and scenarios are needed to explore the role of model structural biases.

This paper is very well written and was a pleasure to read. The simulation results, analysis and scientific arguments are rigorous and sound and, as already mentioned, the findings are quite compelling. The implication is that FRIS is much less susceptible to runaway melting than previous studies had suggested, potentially postponing at least some of the contribution to the most serious scenarios for future sea-level rise.

We are pleased to read these favourable comments, and are glad Dr Asay-Davis agrees that our findings are compelling.

I would recommend this paper for publication after some minor corrections, which I have detailed in my specific comments below. I have also provided some suggested typographic and grammatical corrections.

Specific Comments:

I. 107-108: "However, the second stage requires extremely strong greenhouse forcing, indicating that FRIS melting is unlikely to increase in the 21st century." Above, you rightly point out that previous conclusions about runaway melting under FRIS are based on only one climate scenario that may be outdated. Similarly, in your conclusion and in the

supplement, you are careful to point out some caveats of this work -- the model structural uncertainty that comes from using only one ESM and one coupled ice sheet-ocean model in this study, as well as the idealized scenarios that you adopt from the ESM. Given all of this, it seems like this particular sentence should also convey these caveats somehow. Maybe by reemphasizing that these are the findings of your simulations: "However, in our simulations, the second stage..." Given that this finding is new and has not yet been shown to be robust across multiple models or more realistic scenarios, this caveat seems important to me.

We have update this sentence as suggested (lines 105-108):

However, the second stage requires extremely strong greenhouse forcing in our simulations, indicating that FRIS melting is unlikely to increase in the 21st century.

I. 115: "...a result of increased ice shelf melting upstream...": Could you expound on this a bit? Presumably, the UKESM simulations that provide the forcing and boundary conditions do not include ice shelf melting -- I know that UKESM can simulate ice-shelf cavities but I do not believe this was used in the main CMIP6 submissions. If I am incorrect and ice-shelf melting was part of UKESM, it would be important to include analysis of how realistic these melt rates are (presumably in the supplement). If UKESM doesn't include ice-shelf melting, does this phrase refer to upstream ice shelves in the ÚaMITgcm domain? In which case, which shelves are included? Wouldn't you expect the amount of this upstream melting to be highly dependent on the location of the upstream boundary (how many ice shelves are included)?

Here we are referring to upstream ice shelf melting in the ÚaMITgcm domain. However, ice shelf meltwater is also accounted for in UKESM through a prescribed, spatially-varying freshwater flux derived from observations. The salinity of the ÚaMITgcm domain is therefore not strongly sensitive to the location of the eastern boundary.

In the main text, we have made it more clear that we are referring to upstream ice shelves in ÚaMITgcm (lines 135-138):

This fresher coastal current is partially caused by increased ice shelf melting in the upstream Eastern Weddell sector, within the ÚaMITgcm domain (eastern boundary 30°E).

We have added the following paragraph to Methods to clarify the treatment of ice shelf meltwater in UKESM (lines 457-468)

While this version of UKESM does not include ice shelf cavities, it prescribes a steady three-dimensional coastal freshwater flux representing ice shelf basal melting (Sellar et al., 2020; Mathiot et al., 2017). The spatial distribution of this flux follows the observations of Rignot et al. (2013). UKESM also considers steady iceberg calving through a statistical iceberg model (Marsh et al., 2015). To conserve mass, the sum of the meltwater flux and the calving flux is equal to the modelled preindustrial surface mass balance of Antarctica (not including surface runoff, which is a separate flux to the ocean). This freshwater implicitly enters the MITgcm domain via the lateral boundary conditions. Therefore, the meltwater from ice shelves and icebergs is accounted for both within the ÚaMITgcm domain and in the external forcing from UKESM.

In Section 2 of the supplementary material, we compare the baseline Eastern Weddell ice shelf melting in $\dot{U}a$ MITgcm with observations (lines 80-92):

Upstream of FRIS are the ice shelves of the Eastern Weddell region. These ice shelves are not coupled to $\dot{U}a$, and therefore have fixed geometry, but their basal melt rates still evolve within MITgcm. Transient changes in meltwater from these ice shelves can impact salinity in front of FRIS, as discussed in the main text. The ice shelves with the largest contribution to meltwater are the adjoining Brunt and Riiser-Larsen Ice Shelves (BRLIS) and the adjoining Ekstrom, Jelbart, and Fimbul Ice Shelves (EJFIS). Averaged over the historical simulation, BRLIS has a simulated basal melt flux of 24.5 Gt/y, which is within the observational estimates of Rignot et al. (2013). The simulated melt flux from EJFIS is 56 Gt/y, somewhat above the range given by Rignot et al. (26.8 ± 14 Gt/y). Note that ice shelf meltwater from outside the $\dot{U}a$ MITgcm domain is implicitly taken into account, via the lateral boundary conditions from UKESM (Methods).

Fig. 2: “a) Circulation strength (Sv), defined by the area-averaged absolute value of the barotropic streamfunction in the cavity.” This does not seem like a typical measurement of circulation strength to me. Typically, the maximum (absolute) value of the stream function is used, because this indicates the amount of transport into and out of the cavity. More generally, the difference between the streamfunction values at any two points gives the amount of barotropic transport between those two points. It is not clear that the area average of the barotropic streamfunction has a physical meaning. It can be used to compare the relative strength of circulation between the three simulations but the value in Sv cannot easily be related to the barotropic transport into or out of the cavity. Would it be possible to use the maximum of the barotropic streamfunction instead?

We have changed this timeseries to the maximum absolute value of the streamfunction as suggested. While the short-term variability is larger than before, the smoothed data (11-year running mean) shows a similar pattern: weakened circulation during Stage 1, followed by strengthening circulation as Stage 2 progresses. We have reproduced this panel below (recall that black= π Control, blue=1pctCO2, red=abrupt-4xCO2). The figure caption text has also been updated.

We have also amended the definition of Stage 1 detectability, as this was previously dependent on circulation strength. Further analysis revealed that this metric was very

sensitive to the standard deviation of circulation strength, and much less dependent on the two variables which are more useful from an observational perspective (cavity temperature and basal mass loss). Now that our definition of circulation strength has even higher variability, we have removed it from the calculation of Stage 1 detectability. We have also increased the time window to 20 years, which better captures the timescales of variability in the remaining two variables. The estimate of Stage 1 detectability has therefore changed to 69 years for 1pctCO2 and 14 years for abrupt-4xCO2 (previously it was 75 years and 10 years respectively).

“11-year and 5-year running means respectively”: It wasn't clear to me why different running means are used for these two panels, and it makes comparison between the panels a bit more challenging. Could panel c) be recomputed with an 11-year running mean to keep things more consistent?

We have recomputed panel (c) with an 11-year running mean as suggested, and updated the figure caption accordingly.

Fig. 3: “the 13-month running mean”: I am guessing that an odd number of months was used so that the mean remains centered at the same point in time as the original data. However, using a running mean other than a multiple of 12 months would seem to keep a non-negligible contribution of the seasonal cycle in the plot. Would you please consider replotting these panels with a 12-month running mean?

We have recomputed Figure 3 with a 12-month running mean as suggested. It makes a negligible difference to the plot, but we agree it is more robust in practice.

I. 152-153: “...as weakening sea ice formation allows this region to warm (Supplementary section 5).” Section 5 of the supplement is the animation, which doesn't clearly establish that the warming in the region is caused by weakening sea-ice formation. Perhaps you mean section 4? But section 4 shows only that weakening sea-ice formation is primarily responsible for freshening, not that it is also responsible for warming. Perhaps I missed it but I didn't see the heat budget, equivalent of your salt budget, in the supplement. So it seems plausible but not explicitly established that the weakening sea-ice formation is the cause of the warming. Air-sea heat fluxes or mixing from off the continental shelf also seem like potential sources of warming here. It seems like freshening, rather than warming, is the relevant change for allowing greater MWDW onto the continental shelf and into the Filchner Depression.

This sentence is referring to the Ronne Depression, rather than the Filchner Trough. While the Filchner Trough is the focus of most of the Stage 2 analysis, we also see some modified WDW entering the cavity along its original region of HSSW inflow, the Ronne Depression. This is visible in the animation but is not explored further in other figures, so the reference to Supplementary Section 5 (now Section 3) is appropriate.

We have explained this further in the text to clear up any confusion, and have removed the attribution to sea ice formation (lines 216-219):

Modified WDW also enters the cavity from its original region of inflow, the Ronne Depression (Supplementary Section 3 and Video 1). It follows the existing pathways of HSSW inflow, but the temperature of the water column no longer drops as low as the surface freezing point.

I. 213-215: I appreciated this bit of discussion very much. It sets things up nicely for a community effort to verify the robustness of your findings.

We agree this should be an important priority for the field going forward.

I. 333-334: "Newly opened cells are filled with properties (temperature, salinity, and free surface) of nearest neighbors, preferring vertical neighbors." I was surprised by the choice to prefer vertical neighbors. I have always done the opposite -- extrapolating horizontally where possible and vertically only where necessary, on the assumption that vertical gradients in model index space are sharper than horizontal ones. (This is obviously true for physical gradients, but the aspect ratio of model grid cells already accounts for this to some degree.) After a short adjustment period, I do not think the particular details of the extrapolation strategy matter very much so I do not think this is a major concern, it is simply not an obvious choice to me.

We understand that different groups have made different decisions regarding the extrapolation method for ice-ocean coupling. By preferring vertical neighbours, we ensure that newly opened cells at the ice base are filled with conditions from the ice-ocean boundary layer wherever possible, which we believe is most realistic. Vertical neighbours are only available in situations where existing water columns become deeper (i.e. not in situations of grounding line retreat, where only horizontal neighbours exist). In these cases, a newly opened cell at the ice shelf base will be filled with conditions from the existing boundary layer beneath it. If we were to instead preference horizontal neighbours, this could lead to an unstable water column with warmer, saltier water atop colder, fresher water. The schematic below illustrates the two possibilities.

As Dr Asay-Davis notes, the model quickly adjusts to the new conditions, so the choice of vertical versus horizontal neighbours has little large-scale significance. Indeed, with an

idealised setup of ÚaMITgcm (MISOMIP domain) we tested extreme scenarios in which newly opened cells were uniformly filled with cold (-1.9°C), warm (1°C), fresh (32 psu), or salty (34.7 psu) water. The changes to total ice volume over 100 years ranged from 1% higher to 6-8% lower, which are relatively small changes given the extreme values applied. Furthermore, the extreme cold and fresh scenarios were much closer to the default simulation than the warm and salty scenarios. This is consistent with the expectation that newly opened cells at the ice-ocean interface are generally filled with colder and fresher water, i.e. resembling the boundary layer.

We have added the following sentence to the text to explain our choice (lines 431-433):

This ensures that newly opened cells at the ice base are filled with conditions from the ice-ocean boundary layer wherever possible.

Supplement

I. 117: “Integrated salt fluxes over all depths (Figure S5a)”: I found this phrasing confusing, since most fluxes at layer interfaces or cell edges cancel out when they are integrated over all depths and over a horizontal region. What remains are the fluxes at the surface and those into or out of the region boundaries. So perhaps this could be phrased as, “Integrating salt fluxes over the surface and all depths of the region boundary (Figure S5a)...”

We did indeed calculate fluxes over all cells within the 3D region, but we agree that in practice they will cancel out except at the surface and boundary. We have reworded the text to say simply “Integrated salt fluxes over the continental shelf” (line 119).

Section 5: Thank you for including the animation, it is a nice touch and certainly conveys a lot of information about the simulation that is not available in the figures.

We are glad to hear that the animation was beneficial.

Typographical and grammatical corrections:

Presumably the typesetter will catch many of these but I figured it doesn't hurt to point out what caught my eye. And some of these may just be my personal aesthetic taste, in which case you are free to disregard my suggestions.

We have made all of the following changes as suggested.

I. 1: “...ice shelves which buttress...” should probably be “...ice shelves that buttress...” but could also be “...ice shelves, which buttress...” with a comma.

I. 28-30: Would it be possible to combine the abutting parentheses in these two cases? E.g. “(≈ 1°C; Jacobs et al. 2012)”

I. 47-48: “...climate warming would provoke a similar response to springtime warming...”: I found this phrasing a little confusing. I think you mean, “...climate warming would provoke a response similar to springtime warming...”. But the current phrasing could also be

understood as “...under climate warming, the response to summertime warming would be similar...” (though similar to what exactly would not be clear).

I. 78: The parenthetical statements should be combined, and if one is being especially pedantic “UK” should probably be expanded: “(United Kingdom Earth System Model; Sellar et al., 2019)”.

I. 82-83: Again, parentheticals should be combined: “(Coupled Model Intercomparison Project phase 6; Eyring et al., 2016)”

Fig. 4 caption and I. 445: “parallel piControl simulation”: I find that the term “parallel simulation” tends to invoke the concept of “parallel computing”. I would maybe go with “concurrent piControl simulation” or something similar, or just drop “parallel” entirely.

I. 262-263: Combine the parentheticals: “(Massachusetts Institute of Technology general circulation model; Marshall et al., 1997; Losch, 2008)”.

I. 323: Combine the parentheticals: “(MISOMIP; Asay-Davis et al., 2016)”

I. 360: Combine the parentheticals as in the examples above.

I. 377: A comma is needed between “correction” and “which”

I. 392: A comma is needed between “flux” and “which”

I. 422: The comma between “front” and “at” is not needed.

I. 436: “where” is erroneously indented

Supplement

I. 5: A comma is needed between “(2010)” and “who”

I. 11: A comma is needed between “general” and “as”

I. 17: Combine the parentheticals

I. 43: Combine the parentheticals

I. 50: A comma is needed between “Here” and “we”

I. 51: A comma is needed between “uncertain” and “given”

I. 66: A comma is needed between “Here” and “the”

I. 71-72: “Moholdt et al. (2014)’s estimate”: This is pretty awkward and would be cleaner as “the estimate of Moholdt et al. (2014)”

I. 79: A comma is needed between “tides” and “which”

I. 82: Combine the parentheticals if at all possible.

I. 92: Nested parentheses are not needed: “(a possibility discussed by Naughten et al. 2019)”. If using latex, I believe this is accomplished with `\citealt{}`

I. 112: A comma is needed between “Here” and “we”

We would like to thank Dr Asay-Davis for this thorough and thoughtful review, which led to numerous improvements in our manuscript.

Reviewer 3

The manuscript of Naughten and others simulate with a regional coupled ocean-ice sheet model system the response of the Filchner-Ronne-Ice Shelf (FRIS) under warming climates scenarios. Since FRIS is sustained by ice streams fed by large drainage basins of the Eastern and Western Antarctic Ice Sheets, a disintegration of FRIS would lead to sustained ice loss of these drainage basins. This loss would potentially raise the global sea level substantially.

The model system comprises of the ocean model MITgcm and ice sheet model Úa. The system is driven by corrected forcing extracted from the global earth system model UKESM for three climate scenarios: pi-industrial climate (piControl), and two idealized future scenarios where four-times the pre-industrial atmospheric conditions is reached immediately (abrupt4xCO2) or gradually with a 1% rise per year (1pctCO2). The model shows two distinct responses, which have been described in the literature before as opponent responses. The authors show convincingly that the formerly contradicting reactions are instead interrelated. They argue that former studies may have only reached the so-called “Stage 1” or have advanced directly into the “Stage 2”.

Intermediate warming reduces sea ice formation and the production of High Salinity Shelf Water, which lowers the density gradient between the FRIS cavern and the ambient ocean. This gradient drives the inflow predominately on the western side. Therefore, it reduces the ventilation of the FRIS cavern. Consequently, the mean temperature in the cavern cools, and the integrated basal melting decreases by 20% to 27%. This configuration is labeled as Stage 1. Over time, the temperature drop in FRIS also influences the density gradient on FRIS' eastern side. This is the link between both stages. As soon as this second density gradient drops below a probably model-dependent threshold, the ocean circulation reverses the flow direction. The outflow of cold Ice Shelf Water (ISW) is replaced by Warm Deep Water (WDW) entering the cavern. Consequently, the average temperature and the related basal melting rates skyrocket in FRIS. Ultimately, the grounding line retreats, and ice streams feeding the Filchner-Ronne Ice Shelf accelerates, which raises the sea level.

This study is highly relevant for several reasons.

1. It solves a before not understood difference between independent model simulations.

2. An apparent drop of the basal melting rate is commonly understood as an indication of a stable state. However, this study highlights that it could be a harbinger of a situation where the ice shelf's disintegration becomes more likely or even inevitable.

3. A more nuanced understanding and observation strategy are probably necessary to monitor the stability of some ice shelves surrounding Antarctica.

The study is well organized and written. The figures are of high quality, necessary, and informative. It was a pleasure to read this excellent manuscript.

We are glad to hear of the reviewer's enthusiasm for our work, and agree this paper is highly relevant to the community.

I recommend the publication of the manuscript after a few minor corrections.

General comments

=====

Since changing ocean conditions do not feedback to the atmosphere, which provides the upper boundary conditions for the coupled ocean-ice sheet simulation, the authors have probably used so-called mixed boundary conditions for their setup (Mikolajewicz and Maier-Reimer 1994). These conditions are known to produce artifacts for, at least, the Atlantic Meridional Overturning Circulation (AMOC). Though restricted to a limited region, are your simulation also subject to artifacts because the feedbacks between atmosphere and ocean are not represented?

We do not use traditional mixed boundary conditions with SST restoring as in the paper cited by the reviewer, but rather a bulk flux formulation modulated by an interactive sea ice model. This is a standard design for ocean-ice models such as MITgcm, and has been successful in many studies. However, it is still true that atmosphere-ocean feedbacks are not represented in this class of models. We have therefore added the following text to the discussion of future intercomparison projects in the Implications section (lines 309-313):

Fully coupled models, which simulate the atmosphere, ocean, and ice sheet simultaneously, should be included as they are developed. These models would simulate atmosphere-ocean feedbacks, which may be important to FRIS, but which cannot be captured by ice-ocean models such as ÚaMITgcm.

We have also added a sentence to the Methods section to clarify the use of bulk fluxes (lines 454-456):

The atmospheric fields are used to calculate bulk fluxes within the MITgcm code, which then force the ocean and sea ice models.

Since the UKESM is used as a source for the forcing that drives ÚaMITgcm, it is unclear how UKESM's biases impact the presented conclusion. More specifically: How does the reported temperature bias of 0.5-2°C in the Southern Ocean (Sellar et al. 2019) affect your results? Since it is still common in global climate and earth system models that open-ocean deep convection ventilates the deep Weddell Sea via the rarely occurring Weddell Sea

Polynia, is UKESM also subject to this bias? Since deep convection at the wrong place has a distinct influence on basal melting in FRIS (Naughten et al. 2019). If the UKESM shows the deep ocean ventilation bias, how do the related fluxes that probably drive the regional model ÚaMITgcm, imprint its bias on the presented results?

In fact, UKESM performs very well in the regions used for lateral boundary forcing, particularly in the deep water masses. Below we compare the temperature and salinity in UKESM, time-averaged over the period 1979-2014 in the historical simulation, with the World Ocean Atlas 2018 (WOA18). The output is interpolated to the two boundaries used to force MITgcm.

At the eastern boundary (30°E), UKESM's deep water is within $\pm 0.3^\circ\text{C}$ of WOA18, with virtually no bias in deep salinity. The slope front is slightly deeper in UKESM, appearing as a cold and fresh subsurface bias near the coast, although it should be noted that WOA18 is less reliable in this region due to lack of observations. Further north, UKESM does have a warm and salty near-surface bias (approx $+0.5^\circ\text{C}$ and $+0.25$ psu) but this transient water mass is less likely to persist far into the MITgcm domain, as it is affected by vertical mixing schemes, surface fluxes, and sea ice processes.

Biases are slightly higher at the northern boundary (61°S), but this region is less likely to affect the MITgcm domain due to the direction of the coastal current. Deep water temperature biases are within $\pm 0.5^\circ\text{C}$ of WOA18, with negligible salinity bias. A subsurface warm bias and surface fresh bias are apparent, but again these water masses are unlikely to advect into the MITgcm domain.

We have added a paragraph to Section 2 of Supplementary Information to clarify this point (lines 116-122):

Biases in the deep water masses inherited from UKESM are minimal. Compared to the World Ocean Atlas (Locarnini et al., 2013; Zweng et al., 2013), the UKESM's deep water masses have a temperature bias within $\pm 0.3^{\circ}\text{C}$ at the eastern boundary of MITgcm (Figure S4), and $\pm 0.5^{\circ}\text{C}$ at the northern boundary (not shown). Deep salinity biases are negligible in both regions. Note that the MITgcm domain is more sensitive to conditions at the eastern boundary, due to the direction of the coastal current.

We have also included the figure of the eastern boundary as Figure S4, with the following caption:

Figure S4: (a) Temperature and (b) salinity at 30°E , the eastern boundary of MITgcm. The left column shows output from the UKESM (averaged over years 1979-2014 of the historical simulation), the middle column shows observations from the the World Ocean Atlas (Locarnini et al., 2013; Zweng et al., 2013), and the right column shows the model bias (UKESM minus World Ocean Atlas). All fields are interpolated to the MITgcm grid.

As for open-ocean convection in the Weddell Sea, we agree this has the potential to be important. However, compared to other CMIP models, UKESM is not strongly affected by spurious convection. There is some convection which cycles on and off during the piControl simulation, as indicated by the solid black line in the below figure. Periods of lower sea ice concentration indicate polynyas in the open Weddell Sea. However, we force MITgcm with the latter part of this simulation (years 2880-3060) which is more stable. Also note that there is reason to believe that Weddell Sea polynyas may have been more common in

pre-industrial times (De Lavergne et al., 2014, doi:10.1038/nclimate2132) meaning this periodic convection is not necessarily a bias.

Specific comments

=====

Main document

Page 4, Line 66: I guess your mean: "global climate models have reduced their atmospheric biases."

We have made this change as suggested (lines 65-66).

Page 11, Line 154: You may extent to: "in 1pctCO4, WDW intrusion begins in the last 5 years (model year 145, 5 years after four-times CO2 has been reached) of the main simulation."

We have expanded this sentence to read (lines 220-221):

In 1pctCO2, WDW intrusion begins in the last 5 years of the main simulation (model year 145; Figure 5b).

Page 11, Line 156-157: I do not understand immediately why "the WSW intrusion are imprinted in the atmospheric forcing." Please clarify this point.

We have rephrased this sentence as follows (lines 221-223):

This pulse is repeated each decade of the 50-year extension, indicating that interannual variability in the forcing modulates the initial WDW intrusion.

Page 12, Line 173: I guess you talk about the near-surface or 2m air-temperature and not the actual surface temperature. Could you please clarify?

This is correct: both the model output and observational dataset refer to near-surface air temperature. We have clarified this in the text as follows (lines 259-261):

Observed global mean near-surface air temperature is now $\approx 1^\circ\text{C}$ above pre-industrial, which corresponds to ≈ 30 years of the 1pctCO₂ experiment (Figure 8).

Page 13, Line 181: Here, I guess you mean near-surface and not surface temperature.

We have changed “surface” to “near-surface” as suggested (line 267).

Page 13, Line 191-192: You say “(central Ronne Ice Shelf and east of Berkner Island).” Please refer the reader to Figure 1 or add the label to Figure 4 for Berkner Island.

We have added a reference to Figure 1 as suggested (line 241).

Page 16, Line 233-234: The extraordinary warming of 7°C might be characteristic of the chosen global climate model UKESM. Hence, you may rewrite it to “amount of global warming (e.g., 7°C in UKESM).”

We have clarified that the 7°C figure only refers to the UKESM (lines 328-329):

In order for melt rates to increase, an extraordinary amount of global warming (7°C in the UKESM) is required.

Page 19, Line 281: I refer to the process named “digging,” which may lead to an overly sensitive response. What would happen if you would seal/fill spurious lakes, and how would it influence your results?

We choose to “dig” the bathymetry, rather than filling the problematic regions with land, because the grounding line is easier to discern from observations than the depth of the water column. In other words, for the initial model state beneath the ice shelf, we have more confidence in the observed land-sea mask than in the observed bathymetry. It is not practical to test the impact of sealing or filling these regions in the coupled model, because this would lead to a mismatch with the ice sheet model Úa. Non-negligible regions of the ice sheet would be floating, with none of the dynamic stresses of grounded ice, but would have zero melt rate given by the ocean. This scenario would likely lead to unrealistic behaviour at the grounding line.

Page 21, Line: 325-326: Would the sensitivity increase if you would allow basal melting in partially floating elements proportional to the floating fraction?

The manner in which basal melting should be treated in the vicinity of the grounding line is an ongoing debate in the ice sheet modelling community. Seroussi and Morlighem 2018 (doi:10.5194/tc-12-3085-2018) find that at sufficiently high resolution, the approach of zero melting in partially grounded elements (as we implement in Úa) converges with the approach of scaling melting with the floating fraction (as the reviewer suggests). Based on this analysis, and the high resolution of our Úa configuration, we would not expect a significant sensitivity to the way melting is applied.

However, a closer look at the finite element numerics of Úa indicates that the scaled-melting approach should be avoided for this particular model. Since the finite element shape functions are continuous through the linear elements, any melting within partially grounded elements would necessarily lead to melting in nodes upstream of the grounding line, which is unphysical. We have made this more clear in the manuscript (lines 421-423):

No basal melting is applied in partially grounded elements, to avoid unphysical melting at connected nodes upstream of the grounding line.

Figure 2: You may add to avoid confusion about colored labels: "The color of the time-series, vertical dashed lines, and labels correspond to the scenarios listed in the legend at the bottom."

We have added the following sentence to the caption of Figure 2 (now Figure 5):

The timeseries, dashed vertical lines, and Stage 1/2 labels are all coloured by simulation, as indicated in the legend.

Figure 4: Is it correct that the green line representing the grounding line in piControl appears to be a broken line?

The green line is solid, but some pinning points near the Bailey and Slessor Ice Streams may give the appearance of a dashed line. We have added the following sentence to the caption of Figure 4 (now Figure 7):

Pinning points in the Filchner Ice Shelf appear as isolated green points.

Supplement Material

Page 15, Line 195: Do you mean near-surface air temperature?

This section was merged into the main text, but we have made sure to correctly state "near-surface air temperature" in all instances, including the title and caption of Figure 8.

We would like to thank Reviewer 3 for taking the time to review our manuscript, and for their helpful suggestions which strengthened our analysis.

REVIEWERS' COMMENTS

Reviewer #2 (Remarks to the Author):

Thank you for your thoughtful response and thorough revision of the manuscript. All my concerns have been addressed with these revisions and I believe the paper is ready for publication.

Very nice work!

Reviewer #3 (Remarks to the Author):

Review of Naughten, De Rydt, Rosier, Jenkins, Holland, and Ridley: "Two-timescale response of a large Antarctic ice shelf to climate change" (Nature Communication, Paper: NCOMMS-20-39498-T)

The revised manuscript of Naughten and others describe how the Filchner-Ronne-Ice Shelf (FRIS) reacts to different warming climate scenarios in a regional coupled ocean-ice sheet model system. They drive their coupled model system that is comprised of the ocean model MITgcm and ice sheet model Úa by corrected forcing from the global earth system model UKESM. Three climate scenarios are used: pi-industrial climate (piControl) and two idealized future scenarios where four-times the pre-industrial atmospheric conditions are reached immediately (abrupt4xCO2) or gradually with a 1% rise per year (1pctCO2). As part of the transient response to the applied forcing, their model shows a two-stage response. Former publications have described these as opponent responses, while the authors show that they could be interrelated.

The mechanism of the described response is based on the consequences of the applied warming climate conditions. The ocean freshens in front of the Filchner-Ronne Ice Shelf due to a reduced net sea ice formation and lateral import of modified water masses. These changes alter the circulation-driving density gradient between the continental shelf and the ice shelf cavern. In stage 1, the inflow of warm water masses is hindered (west side), which lows basal ice-shelf melting and, consequently, the melting-driven ice shelf pump. Still, ongoing basal melting reduces the temperature and salinity in the cavern so that the density barrier, preventing inflow on the east side, erodes. Once this east barrier is dragged down, large volumes of warm water enter the cavern and drive skyrocketing basal melting rates.

This study is highly relevant, well organized, and the results are comprehensively discussed. The figures are very helpful in representing the results, and they represent a thorough depiction of model results. The interesting results will certainly impact how we understand the vulnerability of the Filchner-Ronne Ice Shelf.

I recommend the publication of the manuscript without further corrections.

Response to Reviewers

We would like to thank the reviewers for their helpful suggestions on the original version of our manuscript. As indicated by their comments below, no further revisions are required.

Reviewer 2

Thank you for your thoughtful response and thorough revision of the manuscript. All my concerns have been addressed with these revisions and I believe the paper is ready for publication.

Very nice work!

Reviewer 3

The revised manuscript of Naughten and others describe how the Filchner-Ronne-Ice Shelf (FRIS) reacts to different warming climate scenarios in a regional coupled ocean-ice sheet model system. They drive their coupled model system that is comprised of the ocean model MITgcm and ice sheet model Úa by corrected forcing from the global earth system model UKESM. Three climate scenarios are used: pi-industrial climate (piControl) and two idealized future scenarios where four-times the pre-industrial atmospheric conditions are reached immediately (abrupt4xCO₂) or gradually with a 1% rise per year (1pctCO₂). As part of the transient response to the applied forcing, their model shows a two-stage response. Former publications have described these as opponent responses, while the authors show that they could be interrelated.

The mechanism of the described response is based on the consequences of the applied warming climate conditions. The ocean freshens in front of the Filchner-Ronne Ice Shelf due to a reduced net sea ice formation and lateral import of modified water masses. These changes alter the circulation-driving density gradient between the continental shelf and the ice shelf cavern. In stage 1, the inflow of warm water masses is hindered (west side), which lows basal ice-shelf melting and, consequently, the melting-driven ice shelf pump. Still, ongoing basal melting reduces the temperature and salinity in the cavern so that the density barrier, preventing inflow on the east side, erodes. Once this east barrier is dragged down, large volumes of warm water enter the cavern and drive skyrocketing basal melting rates.

This study is highly relevant, well organized, and the results are comprehensively discussed. The figures are very helpful in representing the results, and they represent a thorough depiction of model results. The interesting results will certainly impact how we understand the vulnerability of the Filchner-Ronne Ice Shelf.

I recommend the publication of the manuscript without further corrections.